# TRULY DETERMINISTIC POLICY OPTIMIZATION

## ABSTRACT

In this paper, we present a policy gradient method that avoids exploratory noise injection and performs policy search over the deterministic landscape. By avoiding noise injection all sources of estimation variance can be eliminated in systems with deterministic dynamics (up to the initial state distribution). Since deterministic policy regularization is impossible using traditional non-metric measures such as the KL divergence, we derive a Wasserstein-based quadratic model for our purposes. We state conditions on the system model under which it is possible to establish a monotonic policy improvement guarantee, propose a surrogate function for policy gradient estimation, and show that it is possible to compute exact advantage estimates if both the state transition model and the policy are deterministic. Finally, we describe two novel robotic control environments—one with non-local rewards in the frequency domain and the other with a long horizon (8000 time-steps)— for which our policy gradient method (TDPO) significantly outperforms existing methods (PPO, TRPO, DDPG, and TD3).

Policy Gradient (PG) methods can be broadly characterized by three defining elements: the policy gradient estimator, the regularization measures, and the exploration profile. For gradient estimation, episodic (Williams, 1992), importance-sampling-based (Schulman et al., 2015a), and deterministic (Silver et al., 2014) gradients are some of the most common estimation oracles. As for regularization measures, either an Euclidean distance within the parameter space (Williams, 1992; Silver et al., 2014; Lillicrap et al., 2015), or dimensionally consistent non-metric measures (Schulman et al., 2015a; Kakade & Langford, 2002; Schulman et al., 2017; Kakade, 2002; Wu et al., 2017) have been frequently adapted. Common exploration profiles include Gaussian (Schulman et al., 2015a) and stochastic processes (Lillicrap et al., 2015). These elements form the basis of many model-free and stochastic policy optimization methods successfully capable of learning high-dimensional policy parameters.

Both stochastic and deterministic policy search can be useful in applications. A stochastic policy has the effect of smoothing or filtering the policy landscape, which is desirable for optimization. Searching through stochastic policies has enabled the effective control of challenging environments under a general framework (Schulman et al., 2015a; 2017). The same method could either learn robotic movements or play basic games (1) with minimal domain-specific knowledge, (2) regardless of function approximation classes, and (3) with less human intervention (ignoring reward engineering and hyper-parameter tuning) (Duan et al., 2016). Using stochasticity for exploration, although it imposes approximations and variance, has provided a robust way to actively search for higher rewards. Despite many successes, there are practical environments which remain challenging for current policy gradient methods. For example, non-local rewards (e.g., those defined in the frequency domain), long time horizons, and naturally-resonant environments all occur in realistic robotic systems (Kuo & Golnaraghi, 2002; Meirovitch, 1975; Preumont & Seto, 2008) but can present issues for policy gradient search.

To tackle challenging environments such as these, this paper considers policy gradient methods based on deterministic policies and deterministic gradient estimates, which could offer advantages by allowing the estimation of global reward gradients on long horizons without the need to inject noise into the system for exploration. To facilitate a dimensionally consistent and low-variance deterministic policy search, a compatible policy gradient estimator and a metric measure for regularization should be employed. For gradient estimation we focus on Vine estimators (Schulman et al., 2015a), which can be easily applied to deterministic policies. As a metric measure we use the Wasserstein distance, which can measure meaningful distances between deterministic policy functions that have non-

overlapping supports (in contrast to the Kullback-Liebler (KL) divergence and the Total Variation (TV) distance).

The Wasserstein metric has seen substantial recent application in a variety of machine-learning domains, such as the successful stable learning of generative adversarial models (Arjovsky et al., 2017). Theoretically, this metric has been studied in the context of Lipschitz-continuous Markov decision processes in reinforcement learning (Hinderer, 2005; Ferns et al., 2012). Pirotta et al. (2015) defined a policy gradient method using the Wasserestein distance by relying on Lipschitz continuity assumptions with respect to the policy gradient itself. Furthermore, for Lipschitz-continuous Markov decision processes, Asadi et al. (2018) and Rachelson & Lagoudakis (2010) used the Wasserstein distance to derive model-based value-iteration and policy-iteration methods, respectively. On a more practical note, Pacchiano et al. (2019) utilized Wasserstein regularization for behavior-guided stochastic policy optimization. Moreover, Abdullah et al. (2019) has proposed another robust stochastic policy gradient formulation. Estimating the Wasserstein distance for general distributions is more complicated than typical KL-divergences (Villani, 2008). This fact constitutes and emphasizes the contributions of Abdullah et al. (2019) and Pacchiano et al. (2019). However, for our deterministic observation-conditional policies, closed-form computation of Wasserstein distances is possible without any approximation.

Existing deterministic policy gradient methods (e.g., DDPG and TD3) use *deterministic policies* (Silver et al., 2014; Lillicrap et al., 2015; Fujimoto et al., 2018), meaning that they learn a deterministic policy function from states to actions. However, such methods still use *stochastic search* (i.e., they add stochastic noise to their deterministic actions to force exploration during policy search). In contrast, we will be interested in a method which not only uses *deterministic policies*, but also uses *deterministic search* (i.e., without constant stochastic noise injection). We call this new method *truly deterministic policy optimization* (TDPO) and it may have lower estimation variances and better scalability to long horizons, as we will show in numerical examples.

Scalability to long horizons is one of the most challenging aspects for policy gradient methods that use stochastic search. This issue is sometimes referred to as the *curse of horizon* in reinforcement learning (Liu et al., 2018). General worst-case analyses suggests that the sample complexity of reinforcement learning is exponential with respect to the horizon length (Kakade et al., 2003; Kearns et al., 2000; 2002). Deriving polynomial lower-bounds for the sample complexity of reinforcement learning methods is still an open problem (Jiang & Agarwal, 2018). Lower-bounding the sample complexity of reinforcement learning for long horizons under different settings and simplifying assumptions has been a topic of theoretical research (Dann & Brunskill, 2015; Wang et al., 2020). Some recent work has examined the scalability of importance sampling gradient estimators to long horizons in terms of both theoretical and practical estimator variances (Liu et al., 2018; Kallus & Uehara, 2019; 2020). All in all, long horizons are challenging for all reinforcement learning methods, especially the ones suffering from excessive estimation variance due to the use of stochastic policies for exploration, and our truly deterministic method may have advantages in this respect.

In this paper we focus on continuous-domain robotic environments with reset capability to previously visited states. The main contributions of this work are: (1) we introduce a Deterministic Vine (DeVine) policy gradient estimator which avoids constant exploratory noise injection; (2) we derive a novel deterministically-compatible surrogate function and provide monotonic payoff improvement guarantees; (3) we show how to use the DeVine policy gradient estimator with the Wasserstein-based surrogate in a practical algorithm (TDPO: Truly Deterministic Policy Optimization); (4) we illustrate the robustness of the TDPO policy search process in robotic control environments with non-local rewards, long horizons, and/or resonant frequencies.

## 1 BACKGROUND

**MDP preliminaries.** An infinite-horizon discounted Markov decision process (MDP) is specified by $(\mathcal{S}, \mathcal{A}, P, R, \mu, \gamma)$, where $\mathcal{S}$ is the state space, $\mathcal{A}$ is the action space, $P : \mathcal{S} \times \mathcal{A} \to \Delta(\mathcal{S})$ is the transition dynamics, $R : \mathcal{S} \times \mathcal{A} \to [0, R_{\max}]$ is the reward function, $\gamma \in [0, 1)$ is the discount factor, and $\mu(s)$ is the initial state distribution of interest (where $\Delta(\mathcal{F})$ denotes the set of all probability distributions over $\mathcal{F}$, otherwise known as the Credal set of $\mathcal{F}$). The transition dynamics $P$ is defined as an operator which produces a distribution over the state space for the next state $s' \sim P(s, a)$. The transition dynamics can be easily generalized to take distributions of states or

actions as input (i.e., by having $P$ defined as $P : \Delta(\mathcal{S}) \times \mathcal{A} \rightarrow \Delta(\mathcal{S})$ or $P : \mathcal{S} \times \Delta(\mathcal{A}) \rightarrow \Delta(\mathcal{S})$). We may abuse the notation and replace $\delta_s$ and $\delta_a$ by $s$ and $a$, where $\delta_s$ and $\delta_a$ are the deterministic distributions concentrated at the state $s$ and action $a$, respectively. A policy $\pi : \mathcal{S} \rightarrow \Delta(\mathcal{A})$ specifies a distribution over actions for each state, and induces trajectories from a given starting state $s$ as follows: $s_1 = s$, $a_1 \sim \pi(s_1)$, $r_1 = R(s_1, a_1)$, $s_2 \sim P(s_2, a_2)$, $a_2 \sim \pi(s_2)$, etc. We will denote trajectories as state-action tuples $\tau = (s_1, a_1, s_2, a_2, \dots)$. One can generalize the dynamics (1) by using a policy instead of an action distribution $\mathbb{P}(\mu_s, \pi) := \mathbb{E}_{s \sim \mu_s}[\mathbb{E}_{a \sim \pi(s)}[P(s, a)]]$, and (2) by introducing the $t$-step transition dynamics recursively as $\mathbb{P}^t(\mu_s, \pi) := \mathbb{P}(\mathbb{P}^{t-1}(\mu_s, \pi), \pi)$ with $\mathbb{P}^0(\mu_s, \pi) := \mu_s$, where $\mu_s$ is a distribution over $\mathcal{S}$. The visitation frequency can then be defined as $\rho_\mu^\pi := (1 - \gamma) \sum_{t=1}^{\infty} \gamma^{t-1} \mathbb{P}^{t-1}(\mu, \pi)$. Table 2 of the appendix summarizes all MDP notation.

The value function of $\pi$ is defined as $V^\pi(s) := \mathbb{E}[\sum_{t=1}^{\infty} \gamma^{t-1} r_t \mid s_1 = s; \pi]$. Similarly, one can define $Q^\pi(s, a)$ by conditioning on the first action. The advantage function can then be defined as their difference (i.e. $A^\pi(s, a) := Q^\pi(s, a) - V^\pi(s)$). Generally, one can define the advantage/value of one policy with respect to another using $A^\pi(s, \pi') := \mathbb{E}[Q^\pi(s, a) - V^\pi(s) \mid a \sim \pi'(\cdot|s)]$ and $Q^\pi(s, \pi') := \mathbb{E}[Q^\pi(s, a) \mid a \sim \pi'(\cdot|s)]$. Finally, the payoff of a policy $\eta_\pi := \mathbb{E}[V^\pi(s); s \sim \mu]$ is the average value over the initial states distribution of the MDP.

**Probabilistic and mathematical notations.** Sometimes we refer to $\int f(x)g(x)dx$ integrals as $\langle f, g \rangle_x$ Hilbert space inner products. Assuming that $\zeta$ and $\nu$ are two probabilistic densities, the Kulback-Liebler (KL) divergence is $D_{\mathrm{KL}}(\zeta|\nu) := \langle \zeta(x), \log(\frac{\zeta(x)}{\nu(x)}) \rangle_x$, the Total-Variation (TV) distance is $\mathrm{TV}(\zeta, \nu) =: \frac{1}{2} \langle |\zeta(x) - \nu(x)|, 1 \rangle_x$, and the Wasserstein distance is $W(\zeta, \nu) = \inf_{\gamma \in \Gamma(\zeta, \nu)} \langle \|x - y\|, \gamma(x, y) \rangle_{x,y}$ where $\Gamma(\zeta, \nu)$ is the set of couplings for $\zeta$ and $\nu$. We define $\mathrm{Lip}(f(x, y); x) := \sup_x \|\nabla_x f(x, y)\|_2$ and assume the existence of $\mathrm{Lip}(Q^\pi(s, a); a)$ and $\|\mathrm{Lip}(\nabla_s Q^\pi(s, a); a)\|_2$ constants. Under this notation, the Rubinstein-Kantrovich (RK) duality states that the $|\langle \zeta(x) - \nu(x), f(x) \rangle_x| \leq W(\zeta, \nu) \cdot \mathrm{Lip}(f; x)$ bound is tight for all $f$. For brevity, we may abuse the notation and denote $\sup_s W(\pi_1(\cdot|s), \pi_2(\cdot|s))$ with $W(\pi_1, \pi_2)$ (and similarly for other measures). For parameterized policies, we define $\nabla_\pi f(\pi) := \nabla_\theta f(\pi)$ where $\pi$ is parameterized by the vector $\theta$. Table 1 of the appendix summarizes all these mathematical definitions.

**Policy gradient preliminaries.** The advantage decomposition lemma provides insight into the relationship between payoff improvements and advantages (Kakade & Langford, 2002). That is,

$$\eta_{\pi_2} - \eta_{\pi_1} = \frac{1}{1 - \gamma} \cdot \mathbb{E}_{s \sim \rho_\mu^{\pi_2}}[A^{\pi_1}(s, \pi_2)]. \tag{1}$$

We will denote the current and the candidate next policy as $\pi_1$ and $\pi_2$, respectively. Taking derivatives of both sides with respect to $\pi_2$ at $\pi_1$ yields

$$\nabla_{\pi_2} \eta_{\pi_2} = \frac{1}{1 - \gamma} \left[ \langle \nabla_{\pi_2} \rho_\mu^{\pi_2}(\cdot), A^{\pi_1}(\cdot, \pi_1) \rangle + \langle \rho_\mu^{\pi_1}(\cdot), \nabla_{\pi_2} A^{\pi_1}(\cdot, \pi_2) \rangle \right]. \tag{2}$$

Since $\pi_1$ does not have any advantage over itself (i.e., $A^{\pi_1}(\cdot, \pi_1) = 0$), the first term is zero. Thus, the Policy Gradient (PG) theorem is derived as

$$\nabla_{\pi_2} \eta_{\pi_2} \Big|_{\pi_2 = \pi_1} = \frac{1}{1 - \gamma} \cdot \mathbb{E}_{s \sim \rho_\mu^{\pi_1}}[\nabla_{\pi_2} A^{\pi_1}(s, \pi_2)] \Big|_{\pi_2 = \pi_1}. \tag{3}$$

For policy iteration with function approximation, we assume $\pi_2$ and $\pi_1$ to be parameterized by $\theta_2$ and $\theta_1$, respectively. One can view the PG theorem as a Taylor expansion of the payoff at $\theta_1$.

Conservative Policy Iteration (CPI) (Kakade & Langford, 2002) was one of the early dimensionally consistent methods with a surrogate of the form $\mathcal{L}_{\pi_1}(\pi_2) = \eta_{\pi_1} + \frac{1}{1-\gamma} \cdot \mathbb{E}_{s \sim \rho_\mu^{\pi_1}}[A^{\pi_1}(s, \pi_2)] - \frac{C}{2} \mathrm{TV}^2(\pi_1, \pi_2)$. The $C$ coefficient guarantees non-decreasing payoffs. However, CPI is limited to linear function approximation classes due to the update rule $\pi_{\mathrm{new}} \leftarrow (1 - \alpha)\pi_{\mathrm{old}} + \alpha \pi'$. This lead to the design of the Trust Region Policy Optimization (TRPO) (Schulman et al., 2015a) algorithm.

TRPO exchanged the bounded squared TV distance with the KL divergence by lower bounding it using the Pinsker inequality. This made TRPO closer to the Natural Policy Gradient algorithm(Kakade, 2002), and for Gaussian policies the modified terms had similar Taylor expansions within small trust

regions. Confined trust regions are a stable way of making large updates and avoiding pessimistic coefficients. For gradient estimates, TRPO used Importance Sampling (IS) with a baseline shift:

$$\nabla_{\theta_2} \mathbb{E}_{s \sim \rho_\mu^{\pi_1}} [A^{\pi_1}(s, \pi_2)] \Big|_{\theta_2 = \theta_1} = \nabla_{\theta_2} \mathbb{E}_{\substack{s \sim \rho_\mu^{\pi_1} \\ a \sim \pi_1(\cdot|s)}} \left[ Q^{\pi_1}(s, a) \frac{\pi_2(a|s)}{\pi_1(a|s)} \right] \Big|_{\theta_2 = \theta_1}. \tag{4}$$

While empirical $\mathbb{E}[A^{\pi_1}(s, \pi_2)]$ and $\mathbb{E}[Q^{\pi_1}(s, \pi_2)]$ estimates yield identical variances in principle, the importance sampling estimator in (4) imposes larger variances. Later, Proximal Policy Optimization (PPO) (Schulman et al., 2015b) proposed utilizing the Generalized Advantage Estimation (GAE) method for variance reduction and incorporated first-order smoothing like ADAM (Kingma & Ba, 2014). GAE employed Temporal-Difference (TD) learning (Bhatnagar et al., 2009) for variance reduction. Although TD-learning was not theoretically guaranteed to converge and could add bias, it improved the estimation accuracy.

As an alternative to importance sampling, deterministic policy gradient estimators were also utilized in an actor-critic fashion. Deep Deterministic Policy Gradient (DDPG) (Lillicrap et al., 2015) generalized deterministic gradients by employing Approximate Dynamic Programming (ADP) (Mnih et al., 2015) for variance reduction. Twin Delayed Deterministic Policy Gradient (TD3) (Fujimoto et al., 2018) improved DDPG's approximation to build an even better policy optimization method. Although both methods used deterministic policies, they still performed stochastic search by adding stochastic noise to the deterministic policies to force exploration.

Other lines of stochastic policy optimization were later proposed. Wu et al. (2017) used a Kronecker-factored approximation for curvatures. Haarnoja et al. (2018) proposed a maximum entropy actor-critic method for stochastic policy optimization.

## 2 MONOTONIC POLICY IMPROVEMENT GUARANTEE

We use the Wasserstein metric because it allows the effective measurement of distances between probability distributions or functions with non-overlapping support, such as deterministic policies, unlike the KL divergence or TV distance which are either unbounded or maximal in this case. The physical transition model's smoothness enables the use of the Wasserstein distance to regularize deterministic policies. Therefore, we make the following two assumptions about the transition model:

$$W(\mathbb{P}(\mu, \pi_1), \mathbb{P}(\mu, \pi_2)) \leq L_\pi \cdot W(\pi_1, \pi_2), \tag{5}$$
$$W(\mathbb{P}(\mu_1, \pi), \mathbb{P}(\mu_2, \pi)) \leq L_\mu \cdot W(\mu_1, \mu_2). \tag{6}$$

Also, we make the dynamics stability assumption $\sup \sum_{k=1}^{t} \hat{L}_{\mu, \pi_1, \pi_2}^{(k-1)} \prod_{i=k+1}^{t-1} \tilde{L}_{\mu, \pi_1, \pi_2}^{(i)} < \infty$, with the definitions of the new constants and further discussion of the implications deferred to Section A.5 of the appendix where we also discuss Assumptions 5 and 6 and the existence of $\mathrm{Lip}(Q^\pi(s, a); a)$.

The advantage decomposition lemma can be rewritten as

$$\eta_{\pi_2} = \eta_{\pi_1} + \frac{1}{1-\gamma} \cdot \mathbb{E}_{s \sim \rho_\mu^{\pi_1}} [A^{\pi_1}(s, \pi_2)] + \frac{1}{1-\gamma} \cdot \langle \rho_\mu^{\pi_2} - \rho_\mu^{\pi_1}, A^{\pi_1}(\cdot, \pi_2) \rangle_s. \tag{7}$$

The $\langle \rho_\mu^{\pi_2} - \rho_\mu^{\pi_1}, A^{\pi_1}(\cdot, \pi_2) \rangle$ term has zero gradient at $\pi_2 = \pi_1$, which qualifies it to be crudely called "the second-order term". The theory behind lower-bounding this second-order term with all derivations is left to the appendix. Next, we present the theoretical bottom line.

### 2.1 THE MONOTONIC PAYOFF IMPROVEMENT GUARANTEE

Combining the results of Inequality (30) and Theorems A.5 and A.4 of the appendix leads us to define the regularization terms and coefficients:

$$C_1 := \sup_s \frac{2 \cdot \mathrm{Lip}(Q^{\pi_1}(s, a); a) \cdot \gamma \cdot L_\pi}{(1-\gamma)(1-\gamma L_\mu)}$$

$$C_2 := \sup_s \frac{\| \mathrm{Lip}(\nabla_s Q^{\pi_1}(s, a); a) \|_2 \cdot \gamma \cdot L_\pi}{(1-\gamma)(1-\gamma L_\mu)}$$

$$\mathcal{L}_{WG}(\pi_1, \pi_2; s) := W(\pi_2(a|s), \pi_1(a|s))$$

$$\times \left\| \nabla_{s'} W \left( \frac{\pi_2(a|s') + \pi_1(a|s)}{2}, \frac{\pi_2(a|s) + \pi_1(a|s')}{2} \right) \Big|_{s'=s} \right\|_2. \tag{8}$$

This gives us the novel lower bound for payoff improvement:

$$\mathcal{L}_{\pi_1}^{\text{sup}}(\pi_2) = \eta_{\pi_1} + \frac{1}{1-\gamma} \mathbb{E}_{s \sim \rho_\mu^{\pi_1}}[A^{\pi_1}(s, \pi_2)] - C_1 \cdot \sup_s \left[ \mathcal{L}_{WG}(\pi_1, \pi_2; s) \right]$$

$$- C_2 \cdot \sup_s \left[ W(\pi_2(a|s), \pi_1(a|s))^2 \right]. \tag{9}$$

We have the inequalities $\eta_{\pi_2} \geq \mathcal{L}_{\pi_1}^{\text{sup}}(\pi_2)$ and $\mathcal{L}_{\pi_1}^{\text{sup}}(\pi_1) = \eta_{\pi_1}$. This facilitates the application of Theorem 2.1 as an instance of Minorization-Maximization algorithms (Hunter & Lange, 2004).

**Theorem 2.1.** *Successive maximization of $\mathcal{L}^{\text{sup}}$ yields non-decreasing policy payoffs.*

*Proof.* With $\pi_2 = \arg\max_\pi \mathcal{L}_{\pi_1}^{\text{sup}}(\pi)$, we have $\mathcal{L}_{\pi_1}^{\text{sup}}(\pi_2) \geq \mathcal{L}_{\pi_1}^{\text{sup}}(\pi_1)$. Thus,

$$\eta_{\pi_2} \geq \mathcal{L}_{\pi_1}^{\text{sup}}(\pi_2) \text{ and } \eta_{\pi_1} = \mathcal{L}_{\pi_1}^{\text{sup}}(\pi_1) \implies \eta_{\pi_2} - \eta_{\pi_1} \geq \mathcal{L}_{\pi_1}^{\text{sup}}(\pi_2) - \mathcal{L}_{\pi_1}^{\text{sup}}(\pi_1) \geq 0. \tag{10}$$

$\square$

## 3 SURROGATE OPTIMIZATION AND PRACTICAL ALGORITHM

Successive optimization of $\mathcal{L}_{\pi_1}^{\text{sup}}(\pi_2)$ generates non-decreasing payoffs. However, it is impractical due to the large number of constraints and statistical estimation of maximums. To mitigate this, we take a similar approach to TRPO and optimize for the surrogate

$$\bar{\mathcal{L}}_{\pi_1}(\pi_2) = \eta_{\pi_1} + \frac{1}{1-\gamma} \cdot \mathbb{E}_{s \sim \rho_\mu^{\pi_1}}[A^{\pi_1}(s, \pi_2)] - C_1 \cdot \mathbb{E}_{s \sim \rho_\mu^{\pi_1}} \left[ \mathcal{L}_{WG}(\pi_1, \pi_2; s) \right]$$

$$- C_2 \cdot \mathbb{E}_{s \sim \rho_\mu^{\pi_1}} \left[ W(\pi_2(a|s), \pi_1(a|s))^2 \right]. \tag{11}$$

Although first order stochastic optimization methods can be directly applied to the surrogate defined in Equation (11), second order methods can be more efficient. Since $\mathcal{L}_{WG}(\pi_1, \pi_2; s)$ is the geometric mean of two functions of quadratic order, it is also of quadratic order. However, $\mathcal{L}_{WG}(\pi_1, \pi_2; s)$ may not be twice continuously differentiable. For this, we lower bound $\mathcal{L}_{WG}(\pi_1, \pi_2; s)$ further using the AM-GM inequality and replace it with

$$\mathcal{L}_{G^2}(\pi_1, \pi_2; s) := \left\| \nabla_{s'} W \left( \frac{\pi_2(a|s') + \pi_1(a|s)}{2}, \frac{\pi_2(a|s) + \pi_1(a|s')}{2} \right) \Big|_{s'=s} \right\|_2^2 \tag{12}$$

to form a quadratic regularization term with a definable Hessian matrix (see Section A.8 of the appendix for detailed derivations). This induces our final surrogate which is used for second order optimization:

$$\mathcal{L}_{\pi_1}(\pi_2) = \eta_{\pi_1} + \frac{1}{1-\gamma} \cdot \mathbb{E}_{s \sim \rho_\mu^{\pi_1}}[A^{\pi_1}(s, \pi_2)] - C_1' \cdot \mathbb{E}_{s \sim \rho_\mu^{\pi_1}} \left[ \mathcal{L}_{G^2}(\pi_1, \pi_2; s) \right]$$

$$- C_2' \cdot \mathbb{E}_{s \sim \rho_\mu^{\pi_1}} \left[ W(\pi_2(a|s), \pi_1(a|s))^2 \right]. \tag{13}$$

The coefficients $C_1$ and $C_2$ are dynamics-dependent. For simplicity, we used constant coefficients and a trust region. This yields the Truly Deterministic Policy Optimization (TDPO) as given in Algorithm 1. See the appendix for practical notes on the choice of $C_1$ and $C_2$. Alternatively, one could adopt processes similar to Schulman et al. (2015a) where the IS-based advantage estimator used a line search for proper step size selection, or the adaptive penalty coefficient setting in Schulman et al. (2017). We plan to consider such approaches in future work.

### 3.1 ON THE INTERPRETATION OF THE SURROGATE FUNCTION

For deterministic policies, the squared Wasserstein distance $W(\pi_2(a|s), \pi_1(a|s))^2$ degenerates to the Euclidean distance over the action space. Any policy defines a sensitivity matrix at a given state $s$, which is the Jacobian matrix of the policy output with respect to $s$. The policy sensitivity

---
**Algorithm 1** Truly Deterministic Policy Optimization (TDPO)

---
**Require:** The squared Wasserestein regularization coefficient $C_2'$, The secondary regularization coefficient ratio $C_1'/C_2'$, and a trust region radius $\delta_{\max}$.
**Require:** Initial policy $\pi_0$.
**Require:** Advantage estimator and sample collector oracle $\mathbb{A}^\pi$.
  1: **for** $k = 1, 2, \ldots$ **do**
  2:     Collect trajectories and construct the advantage estimator oracle $\mathbb{A}^{\pi_k}$.
  3:     Compute the policy gradient $g$ at $\theta_k : g \leftarrow \nabla_{\theta'} \mathbb{A}^{\pi_k}(\pi')|_{\pi'=\pi_k}$
  4:     Construct a surrogate Hessian vector product oracle $v \to H \cdot v$ such that for $\theta' = \theta_k + \delta\theta$,

$$\mathbb{E}_{s \sim \rho_\mu^{\pi_k}} \left[ W(\pi'(a|s), \pi_k(a|s))^2 \right] + \frac{C_1'}{C_2'} \mathbb{E}_{s \sim \rho_\mu^{\pi_k}} \left[ \mathcal{L}_{G^2}(\pi', \pi_k; s) \right] = \frac{1}{2} \delta\theta^T H \delta\theta + \text{h.o.t.}, \quad (14)$$

   where h.o.t. denotes higher order terms in $\delta\theta$.
  5:     Find the optimal update direction $\delta\theta^* = H^{-1}g$ using the Conjugate Gradient algorithm.
  6:     Determine the best step size $\alpha^*$ within the trust region:

$$\alpha^* = \arg\max_\alpha g^T(\alpha\delta\theta^*) - \frac{C_2'}{2}(\alpha\delta\theta^*)^T H(\alpha\delta\theta^*)$$

$$\text{s.t.} \quad \frac{1}{2}(\alpha^*\delta\theta^*)^T H(\alpha^*\delta\theta^*) \leq \delta_{\max}^2 \quad (15)$$

  7:     Update the policy parameters: $\theta_{k+1} \leftarrow \theta_k + \alpha^*\delta\theta^*$.
  8: **end for**

---

term $\mathcal{L}_{G^2}(\pi_1, \pi_2; s)$ is essentially the squared Euclidean distance over the action-to-observation Jacobian matrix elements. In other words, our surrogate prefers to step in directions where the action-to-observation sensitivity is preserved within updates.

Although our surrogate uses a metric distance instead of the traditional non-metric measures for regularization, we do not consider this sole replacement a major contribution. The squared Wasserestein distance and the KL divergence of two identically-scaled Gaussian distributions are the same up to a constant (i.e., $D_{\text{KL}}(\mathcal{N}(m_1, \sigma)\|\mathcal{N}(m_2, \sigma)) = W(\mathcal{N}(m_1, \sigma), \mathcal{N}(m_2, \sigma))^2/2\sigma^2$). On the other hand, our surrogate's compatibility with deterministic policies makes it a valuable asset for our policy gradient algorithm; both $W(\pi_2(a|s), \pi_1(a|s))^2$ and $\mathcal{L}_{G^2}(\pi_1, \pi_2; s)$ can be evaluated for two deterministic policies $\pi_1$ and $\pi_2$ numerically without resorting to any approximations to overcome singularities.

## 4 MODEL-FREE ESTIMATION OF POLICY GRADIENT

The DeVine advantage estimator is formally defined in Algorithm 2. Unlike DDPG and TD3, the DeVine estimator allows our method to perform *deterministic search* by not consistently injecting noise in actions for exploration. Essentially, DeVine rolls out a trajectory and computes the values of each state. Since the transition dynamics and the policy are deterministic, these values are exact. Then, it picks a perturbation state $s_t$ according to the visitation frequencies using importance sampling. A state-reset to $s_t$ is made, a $\sigma$-perturbed action is applied for a single time-step, followed by $\pi_1$ policy. This exactly produces $Q^{\pi_1}(s_t, a_t + \sigma)$. Then, $A^{\pi_1}(s_t, a_t + \sigma)$ can be computed by subtracting the value baseline. Finally, $A^{\pi_1}(s_t, a_t) = 0$ and $A^{\pi_1}(s_t, a_t + \sigma)$ define a two-point linear $A^{\pi_1}(s_t, a)$ model with respect to the action. Parallelization can be used to have as many states of the first roll-out included in the estimator as desired. The parameter $\sigma$ acts as an exploration parameter and a finite difference to establish derivatives. While $\sigma \simeq 0$ can produce exact gradients, larger $\sigma$ can build stabler interpolations.

Under deterministic dynamics and policies, if the DeVine oracle samples each dimension at each time-step exactly once then in the limit of small $\sigma$ it can produce exact advantages, as stated in Theorem 4.1, whose proof is deferred to the appendix.

**Theorem 4.1.** *Assume a finite horizon MDP with both deterministic transition dynamics $P$ and initial distribution $\mu$, with maximal horizon length of $H$. Define $K = H \cdot \dim(\mathcal{A})$, a uniform $\nu$, and*

---

**Algorithm 2** Deterministic Vine (DeVine) Policy Advantage Estimator

---

**Require:** The number of parallel workers $K$

**Require:** A policy $\pi$, an exploration policy $q$, discrete time-step distribution $\nu(t)$, initial state distribution $\mu(s)$, and the discount factor $\gamma$.

1: Sample an initial state $s_0$ from $\mu$, and then roll out a trajectory $\tau = (s_0, a_0, s_1, a_1, \cdots)$ using $\pi$.
2: **for** $k = 1, 2, \cdots, K$ **do**
3:      Sample the integer number $t = t_k$ from $\nu$.
4:      Compute the value $V^{\pi_1}(s_t) = \sum_{i=t}^{\infty} \gamma^{t-i} R(s_i, a_i)$.
5:      Reset the initial state to $s_t$, sample the first action $a'_t$ according to $q(\cdot|s_t)$, and use $\pi$ for the rest of the trajectory. This will create $\tau' = (s_t, a'_t, s'_{t+1}, a'_{t+1}, \cdots)$.
6:      Compute the value $Q^{\pi_1}(s_t, a'_t) = \sum_{i=t}^{\infty} \gamma^{t-i} R(s'_i, a'_i)$.
7:      Compute the advantage $A^{\pi_1}(s_t, a'_t) = Q^{\pi}(s_t, a'_t) - V^{\pi}(s_t)$.
8: **end for**
9: Define $\mathbb{A}^{\pi_1}(\pi_2) := \frac{1}{K} \sum_{k=1}^{K} \frac{\dim(\mathcal{A}) \cdot \gamma^{t_k}}{\nu(t_k)} \cdot \frac{(\pi_2(s) - a_{t_k})^T (a'_{t_k} - a_{t_k})}{(a'_{t_k} - a_{t_k})^T (a'_{t_k} - a_{t_k})} \cdot A^{\pi_1}(s_{t_k}, a'_{t_k})$.
10: Return $\mathbb{A}^{\pi_1}(\pi_2)$ and $\nabla_{\pi_2} \mathbb{A}^{\pi_1}(\pi_2)$ as unbiased estimators for $E_{s \sim \rho_\mu^{\pi_1}}[A^{\pi_1}(s, \pi_2)]$ and PG, respectively.

---

$q(s; \sigma) = \pi_1(s) + \sigma \mathbf{e}_j$ *in Algorithm 2 with* $\mathbf{e}_j$ *being the* $j^{th}$ *basis element for* $\mathcal{A}$*. If the* $(j, t_k)$ *pairs are sampled to exactly cover* $\{1, \cdots, \dim(\mathcal{A})\} \times \{1, \cdots, H\}$*, then we have*

$$\lim_{\sigma \to 0} \nabla_{\pi_2} \mathbb{A}^{\pi_1}(\pi_2)\big|_{\pi_2 = \pi_1} = \nabla_{\pi_2} \eta_{\pi_2}\big|_{\pi_2 = \pi_1}. \tag{16}$$

Theorem 4.1 provides a guarantee for recovering the exact policy gradient if the initial state distribution was deterministic and all time-steps of the trajectory were used to branch vine trajectories. Although this theorem sets the stage for computing a fully deterministic gradient, stochastic approximation can be used in Algorithm 2 by randomly sampling a small set of states for advantage estimation. In other words, Theorem 4.1 would use $\nu$ to deterministically sample all trajectory states, whereas this is not a practical requirement for Algorithm 2 and the gradients are still unbiased if a random set of vine branches is used.

The DeVine estimator can be advantageous in at least two scenarios. First, in the case of rewards that cannot be decomposed into summations of immediate rewards. For example, overshoot penalizations or frequency-based rewards as used in robotic systems are non-local. DeVine can be robust to non-local rewards as it is insensitive to whether the rewards were applied immediately or after a long period. Second, DeVine can be an appropriate choice for systems that are sensitive to the injection of noise, such as high-bandwidth robots with natural resonant frequencies. In such cases, using white (or colored) noise for exploration can excite these resonant frequencies and cause instability, making learning difficult. DeVine avoids the need for constant noise injection.

## 5 EXPERIMENTS

The next two subsections show challenging robotic control tasks including frequency-based non-local rewards, long horizons, and sensitivity to resonant frequencies. See Section A.11 of the appendix for a comparison on traditional gym environments. TDPO works similar or slightly worse on these traditional gym environments as they seem to be well-suited for stochastic exploration.

### 5.1 AN ENVIRONMENT WITH NON-LOCAL REWARDS [1]

The first environment that we consider is a simple pendulum. The transition function is standard—the states are joint angle and joint velocity, and the action is joint torque. The reward function is non-standard—rather than define a local reward in the time domain with the goal of making the pendulum

---

[1] Non-local rewards are reward functions of the entire trajectory whose payoffs cannot be decomposed into the sum of terms such as $\eta = \sum_t f_t(s_t, a_t)$, where functions $f_t$ only depend on nearby states and actions. An example non-local reward is one that depends on the Fourier transform of the complete trajectory signal.

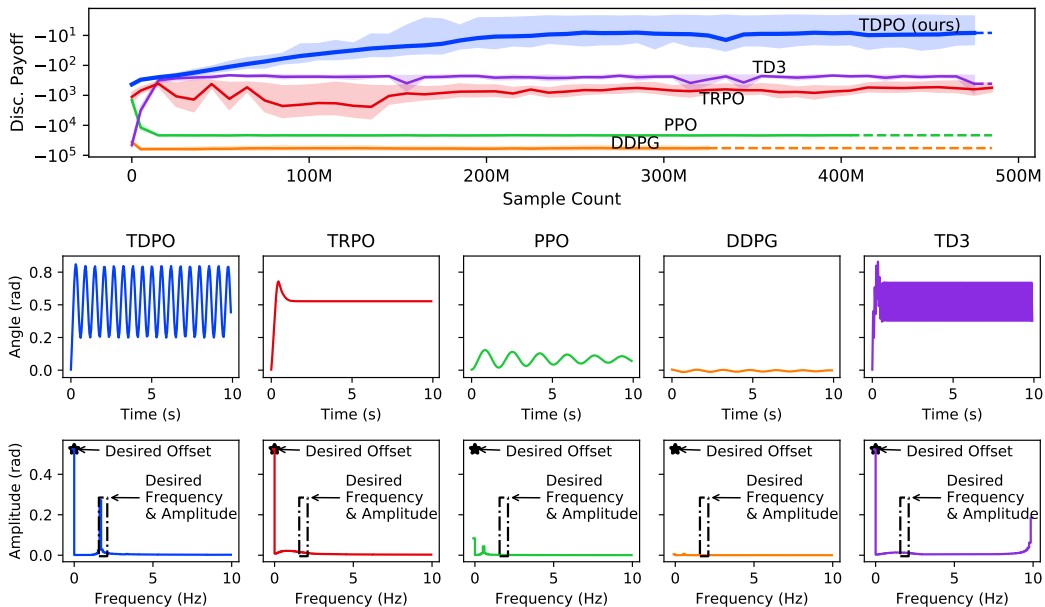

Figure 1: Results for the simple pendulum with non-local rewards. Upper panel: training curves with empirical discounted payoffs. Lower panels: trajectories in both the time domain and frequency domain, showing target values of oscillation frequency, amplitude, and offset.

stand upright (for example), we define a non-local reward in the frequency domain with the goal of making the pendulum oscillate with a desired frequency and amplitude about a desired offset. In particular, we compute this non-local reward by taking the Fourier transform of the joint angle signal over the entire trajectory and by penalizing differences between the resulting power spectrum and a desired power spectrum. We apply this non-local reward at the last time step of the trajectory. Implementation details and similar results for more pendulum variants are left to the appendix.

Figure 1 shows training curves for TDPO (our method) as compared to TRPO, PPO, DDPG, and TD3. These results were averaged over 25 experiments in which the desired oscillation frequency was 1.7 Hz (different from the pendulum's natural frequency of 0.5 Hz), the desired oscillation amplitude was 0.28 rad, and the desired offset was 0.52 rad. Figure 1 also shows trajectories obtained by the best agents from each method. TDPO (our method) was able to learn high-reward behavior and to achieve the desired frequency, amplitude, and offset. TRPO was able to learn the correct offset but did not produce any oscillatory behavior. TD3 also learned the correct offset, but could not produce desirable oscillations. PPO and DDPG failed to learn any desired behavior.

## 5.2 An Environment with Long Horizon and Resonant Frequencies[2]

The second environment that we consider is a single leg from a quadruped robot (Park et al., 2017). This leg has two joints, a "hip" and a "knee," about which it is possible to exert torques. The hip is attached to a slider that confines motion to a vertical line above flat ground. We assume the leg is dropped from some height above the ground and the task is to recover from this drop and to stand upright at rest after impact. States given to the agent are the angle and velocity of each joint (slider position and velocity are hidden), and actions are the joint torques. The reward function penalizes difference from an upright posture, slipping or chattering at the contact between the foot and the

---

[2]Resonant frequencies are a concept from control theory. In the frequency domain, signals of certain frequencies are excited more than others when applied to a system. This is captured by the frequency-domain transfer function of the system, which may have a peak of magnitude greater than one. The resonant frequency is the frequency at which the frequency-domain transfer function has the highest amplitude. Common examples of systems with a resonant frequency include the undamped pendulum, which oscillates at its natural frequency, and RLC circuits which have characteristic frequencies at which they are most excitable. See Chapter 8 of Kuo & Golnaraghi (2002) for more information.

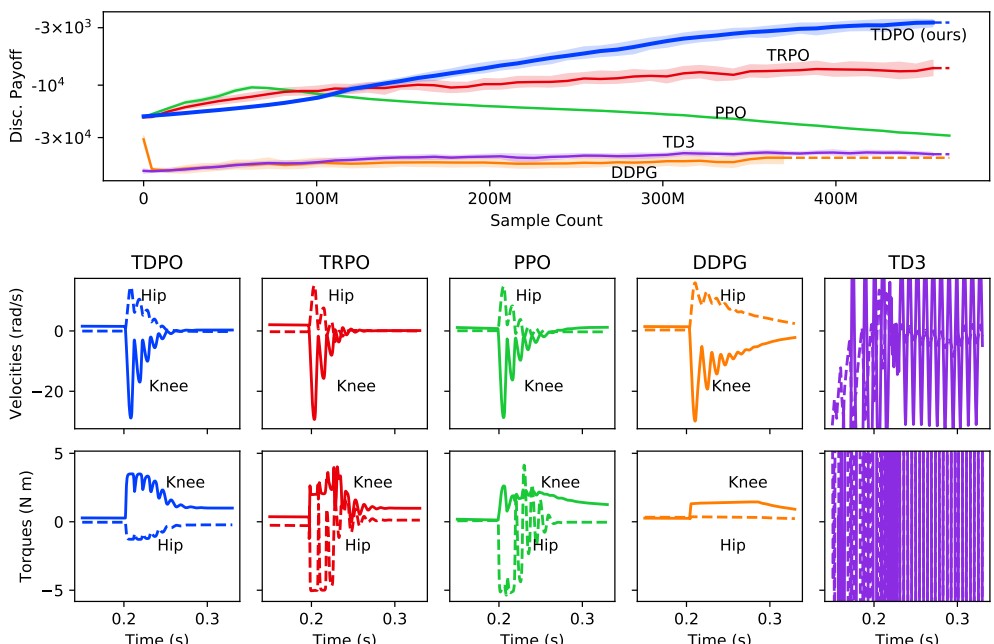

Figure 2: Results for the leg environment with a long horizon and resonant frequencies due to ground compliance. Upper panel: training curves with empirical discounted payoffs. Lower panel: partial trajectories, restricted to times shortly before and after impact with the ground. Note the oscillations at about 100 Hz that appear just after the impact at 0.2 s—these oscillations are evidence of a resonant frequency.

ground, non-zero joint velocities, and steady-state joint torque deviations. We use MuJoCo for simulation (Todorov et al., 2012), with high-fidelity models of ground compliance, motor nonlinearity, and joint friction. The control loop rate is 4000 Hz and the rollout length is 2 s, resulting in a horizon of 8000 steps. Implementation details are left to the appendix.

Figure 2 shows training curves for TDPO (our method) as compared to TRPO, PPO, DDPG and TD3. These results were averaged over 75 experiments. A discount factor of $\gamma = 0.99975$ was chosen for all methods, where $(1 - \gamma)^{-1}$ is half the trajectory length. Similarly, the GAE factors for PPO and TRPO were scaled up to 0.99875 and 0.9995, respectively, in proportion to the trajectory length. Figure 2 also shows trajectories obtained by the best agents from each method. TDPO (our method) was able to learn high-reward behavior. TRPO, PPO, DDPG, and TD3 were not.

We hypothesize that the reason for this difference in performance is that TDPO better handles the combination of two challenges presented by the leg environment—an unusually long time horizon (8000 steps) and the existence of a resonant frequency that results from compliance between the foot and the ground (note the oscillations at a frequency of about 100 Hz that appear in the trajectories after impact). Both high-speed control loops and resonance due to ground compliance are common features of real-world legged robots to which TDPO seems to be more resilient.

# 6 Discussion

This article proposed a deterministic policy gradient method (TDPO: Truly Deterministic Policy Optimization) based on the use of a deterministic Vine (DeVine) gradient estimator and the Wasserstein metric. We proved monotonic payoff guarantees for our method, and proposed a novel surrogate for policy optimization. We showed numerical evidence for superior performance with non-local rewards defined in the frequency domain and a realistic long-horizon resonant environment. This method enables applications of policy gradient to customize frequency response characteristics of agents. Future work should include the addition of a regularization coefficient adaptation process for the $C_1$ and $C_2$ parameters in the TDPO algorithm.

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

# A  APPENDIX

## A.1  TABLES OF NOTATION

The same mathematical definitions and notations used in the paper were re-introduced and summarized in two tables; Table 1 describes the mathematical functions and operators used throughout the paper, and Table 2 describes the notations needed to define the Markov Decision Process (MDP). The tables consist of two columns; one showing or defining the notation, and the other includes the name in which the same notation was called in the paper.

| Name | Mathematical Definition or Description |
|---|---|
| Value function | $V^\pi(s) := \frac{1}{1-\gamma} \mathbb{E}_{\substack{s_t \sim \rho^\pi_\mu \\ a_t \sim \pi(s_t)}} [R(s_t, a_t)]$ $= \mathbb{E}[\sum_{t=1}^\infty \gamma^{t-1} R(s_t, a_t)\|s_1 = s, a_t \sim \pi(s_t), s_{t+1} \sim P(s_t, a_t)].$ |
| Q-Value function | $Q^\pi(s, a) := R(s, a) + \gamma \cdot \mathbb{E}_{s' \sim P(s,a)}[V^\pi(s')]$ |
| Advantage function | $A^\pi(s, a) := Q^\pi(s, a) - V^\pi(s).$ |
| Advantage function | $A^\pi(s, \pi') := \mathbb{E}_{a \sim \pi'(s)}[A^\pi(s, a)].$ |
| Arbitrary functions | $f$ and $g$ are arbitrary functions used next. |
| Arbitrary distributions | $\nu$ and $\zeta$ are arbitrary probability distributions used next. |
| Hilbert inner product | $\langle f, g \rangle_x := \int f(x)g(x)\mathrm{d}x$ |
| Kulback-Liebler (KL) divergence | $D_{\mathrm{KL}}(\zeta\|\nu) := \langle \zeta(x), \log(\frac{\zeta(x)}{\nu(x)}) \rangle_x = \int \zeta(x) \log(\frac{\zeta(x)}{\nu(x)})\mathrm{d}x$ |
| Total Variation (TV) distance | $\mathrm{TV}(\zeta, \nu) := \frac{1}{2}\langle \|\zeta(x) - \nu(x)\|, 1 \rangle_x = \frac{1}{2}\int \|\zeta(x) - \nu(x)\|\mathrm{d}x.$ |
| Coupling set | $\Gamma(\zeta, \nu)$ is the set of couplings for $\zeta$ and $\nu$. |
| Wasserstein distance | $W(\zeta, \nu) = \inf_{\gamma \in \Gamma(\zeta,\nu)} \langle \|x - y\|, \gamma(x, y) \rangle_{x,y}.$ |
| Policy Wasserstein distance | $W(\pi_1, \pi_2) := \sup_{s \in \mathcal{S}} W(\pi_1(\cdot\|s), \pi_2(\cdot\|s)).$ |
| Lipschitz Constant | $\mathrm{Lip}(f(x, y); x) := \sup_x \|\nabla_x f(x, y)\|_2.$ |
| Rubinstein-Kantrovich (RK) duality | $\|\langle \zeta(x) - \nu(x), f(x) \rangle_x\| \leq W(\zeta, \nu) \cdot \mathrm{Lip}(f; x).$ |

Table 1: The mathematical notations used throughout the paper.

| Mathematical Notation | Name and Description |
|---|---|
| $\mathcal{S}$ | This is the state space of the MDP. |
| $\mathcal{A}$ | This is the action space of the MDP. |
| $\gamma$ | This is the discount factor of the MDP. |
| $R : \mathcal{S} \times \mathcal{A} \to \mathbb{R}$ | This is the reward function of the MDP. |
| $\mu$ | This is the initial state distribution of the MDP over the state space. |
| $\Delta$ | $\Delta(\mathcal{F})$ is the set of all probability distributions over the arbitrary set $\mathcal{F}$ (otherwise known as the Credal set of $\mathcal{F}$). |
| $\pi$ | In general, $\pi$ denotes the policy of the MDP. However, the output argument type could vary in the text. See the next lines. |
| $\pi : \mathcal{S} \to \Delta(\mathcal{A})$ | Given a state $s \in \mathcal{S}$, $\pi(s)$ and $\pi(\cdot|s)$ denote the action distribution suggested by the policy $\pi$.

In other words, $a \sim \pi(s)$ and $a \sim \pi(\cdot|s)$. |
| $\pi_{\mathrm{det}} : \mathcal{S} \to \mathcal{A}$ | For a deterministic policy $\pi_{\mathrm{det}}$, the unique action $a$ suggested by the policy given the state $s$ can be denoted by $\pi(s)$ specially.

In other words, $a = \pi_{\mathrm{det}}(s)$. |
| $\Pi$ | $\Pi$ is the set of all policies (i.e., $\forall \pi : \pi \in \Pi$). |
| $P$ | In general, $P$ denotes the transition dynamics model of the MDP. However, the input argument types could vary throughout the text. See the next lines for more clarification. |
| $P : \mathcal{S} \times \mathcal{A} \to \Delta(\mathcal{S})$ | Given a particular state $s$ and action $a$, $P(s, a)$ will be the next state distribution of the transition dynamics (i.e. $s' \sim P(s, a)$ where $s'$ denotes the next state after applying $s, a$ to the transition $P$). |
| $P : \Delta(\mathcal{S}) \times \mathcal{A} \to \Delta(\mathcal{S})$ | This is a generalization of the transition dynamics to accept state distributions as input. In other words, $P(\nu_s, a) := \mathbb{E}_{s \sim \nu_s}[P(s, a)]$. |
| $P : \mathcal{S} \times \Delta(\mathcal{A}) \to \Delta(\mathcal{S})$ | This is a generalization of the transition dynamics to accept action distributions as input. In other words, $P(s, \nu_a) := \mathbb{E}_{a \sim \nu_a}[P(s, a)]$. |
| $\mathbb{P} : \Delta(\mathcal{S}) \times \Pi \to \Delta(\mathcal{S})$ | This is a generalization of the transition dynamics to accept a state distribution and a policy as input. Given an arbitrary state distribution $\nu_s$ and a policy $\pi$, and $\mathbb{P}(\nu_s, \pi)$ will be the next state distribution given that the state is sampled from $\nu_s$ and the action is sampled from the $\pi(s)$ distribution.

In other words, we have $\mathbb{P}(\nu_s, \pi) := \mathbb{E}_{\substack{s \sim \nu_s \\ a \sim \pi(s)}}[P(s, a)]$. |
| $\mathbb{P}^t : \Delta(\mathcal{S}) \times \Pi \to \Delta(\mathcal{S})$ | This is the $t$-step transition dynamics generalization. Given an arbitrary state distribution $\nu_s$ and a policy $\pi$ and non-negative integer $t$, one can define $\mathbb{P}^t$ recursively as $\mathbb{P}^0(\nu_s, \pi) := \nu_s$ and $\mathbb{P}^t(\nu_s, \pi) := \mathbb{P}(\mathbb{P}^{t-1}(\nu_s, \pi), \pi)$. |
| $\rho_\mu^\pi$ | The discounted visitation frequency $\rho_\mu^\pi$ is a distribution over $\mathcal{S}$, and can be defined as $\rho_\mu^\pi := (1 - \gamma) \sum_{t=0}^\infty \gamma^t \mathbb{P}^t(\mu, \pi)$. |

Table 2: The MDP notations used throughout the paper.

## A.2 BOUNDING $W(\mathbb{P}^t(\mu, \pi_1), \mathbb{P}^t(\mu, \pi_2))$

To review, the dynamical smoothness assumptions were

$$W(\mathbb{P}(\mu, \pi_1), \mathbb{P}(\mu, \pi_2)) \leq L_\pi \cdot W(\pi_1, \pi_2),$$
$$W(\mathbb{P}(\mu_1, \pi), \mathbb{P}(\mu_2, \pi)) \leq L_\mu \cdot W(\mu_1, \mu_2).$$

The following lemma states that these two assumptions are equivalent to a more concise assumption. This will be used to bound the $t$-step visitation distance and prove Lemma A.2.

**Lemma A.1.** *Assumptions (5) and (6) are equivalent to having*

$$W(\mathbb{P}(\mu_1, \pi_1), \mathbb{P}(\mu_2, \pi_2)) \leq L_\mu \cdot W(\mu_1, \mu_2) + L_\pi \cdot W(\pi_1, \pi_2). \qquad (17)$$

*Proof.* To prove the $(5), (6) \Rightarrow (17)$ direction, the triangle inequality for the Wasserstein distance gives

$$W(\mathbb{P}(\mu_1, \pi_1), \mathbb{P}(\mu_2, \pi_2)) \leq W(\mathbb{P}(\mu_1, \pi_1), \mathbb{P}(\mu_2, \pi_1)) + W(\mathbb{P}(\mu_2, \pi_1), \mathbb{P}(\mu_2, \pi_2)) \qquad (18)$$

and using (5), (6), and (18) then implies

$$W(\mathbb{P}(\mu_1, \pi_1), \mathbb{P}(\mu_2, \pi_2)) \leq L_\mu \cdot W(\mu_1, \mu_2) + L_\pi \cdot W(\pi_1, \pi_2). \qquad (19)$$

The other direction is trivial. $\qquad \square$

**Lemma A.2.** *Under Assumptions (5) and (6) we have the bound*

$$W(\mathbb{P}^t(\mu, \pi_1), \mathbb{P}^t(\mu, \pi_2)) \leq L_\pi \cdot (1 + L_\mu + \cdots + L_\mu^{t-1}) \cdot W(\pi_1, \pi_2), \qquad (20)$$

*where $\mathbb{P}^t(\mu, \pi)$ denotes the state distribution after running the MDP for $t$ time-steps with the initial state distribution $\mu$ and policy $\pi$.*

*Proof.* For $t = 1$, the lemma is equivalent to Assumption (5). This paves the way for the lemma to be proved using induction. The hypothesis is

$$W(\mathbb{P}^{t-1}(\mu, \pi_1), \mathbb{P}^{t-1}(\mu, \pi_2)) \leq L_\pi \cdot (1 + L_\mu + \cdots + L_\mu^{t-2}) \cdot W(\pi_1, \pi_2), \qquad (21)$$

and for the induction step we write

$$W(\mathbb{P}^t(\mu, \pi_1), \mathbb{P}^t(\mu, \pi_2)) = W(\mathbb{P}(\mathbb{P}^{t-1}(\mu, \pi_1), \pi_1), \mathbb{P}(\mathbb{P}^{t-1}(\mu, \pi_2), \pi_2)). \qquad (22)$$

Using Assumption (17), which according to Lemma A.1 is equivalent to Assumptions (5) and (6), we can combine (21) and (22) into

$$W(\mathbb{P}^t(\mu, \pi_1), \mathbb{P}^t(\mu, \pi_2)) \leq L_\pi \cdot W(\pi_1, \pi_2) + L_\mu \cdot W(\mathbb{P}^{t-1}(\mu, \pi_1), \mathbb{P}^{t-1}(\mu, \pi_2)). \qquad (23)$$

Thus, by applying the induction Hypothesis (21), we have

$$W(\mathbb{P}^t(\mu, \pi_1), \mathbb{P}^t(\mu, \pi_2)) \leq L_\pi \cdot W(\pi_1, \pi_2) + L_\mu \cdot L_\pi \cdot (1 + L_\mu + \cdots + L_\mu^{t-2}) \cdot W(\pi_1, \pi_2), \qquad (24)$$

which can be simplified into the lemma statement (i.e., Inequality (20)). $\qquad \square$

## A.3 BOUNDING $W(\rho_\mu^{\pi_1}, \rho_\mu^{\pi_2})$

Lemma A.2 suggests making the $\gamma L_\mu < 1$ assumption and paves the way for Theorem A.4. The $\gamma L_\mu < 1$ assumption is overly restrictive and unnecessary, but makes the rest of the proof easier to follow. This assumption can be relaxed by a general transition dynamics stability assumption which is discussed in more detail later at section A.5.3, and an equivalent $\gamma \bar{L}_\mu < 1$ assumption is introduced to replace $\gamma L_\mu < 1$.

First, we need to introduce Lemma A.3 first, which will be used in the proof of Theorem A.4.

**Lemma A.3.** *The Wasserstein distance between linear combinations of distributions can be bounded as $W(\beta \cdot \mu_1 + (1 - \beta) \cdot \nu_1, \beta \cdot \mu_2 + (1 - \beta) \cdot \nu_2) \leq \beta \cdot W(\mu_1, \mu_2) + (1 - \beta) \cdot W(\nu_1, \nu_2)$.*

*Proof.* Plugging $\gamma = \beta \cdot \gamma_{(\mu_1, \mu_2)} + (1 - \beta) \cdot \gamma_{(\nu_1, \nu_2)}$ in the Wasserstein definition yields the result. $\qquad \square$

**Theorem A.4.** *Assuming (5), (6), and $\gamma L_\mu < 1$, we have the inequality*

$$W(\rho_\mu^{\pi_1}, \rho_\mu^{\pi_2}) \leq \frac{\gamma L_\pi}{1 - \gamma L_\mu} \cdot W(\pi_1, \pi_2). \tag{25}$$

*Proof.* Using Lemma A.3 and the definition of $\rho_\mu^\pi$, we can write

$$W(\rho_\mu^{\pi_1}, \rho_\mu^{\pi_2}) \leq (1 - \gamma) \sum_{t=0}^{\infty} \gamma^t \cdot W(\mathbb{P}^t(\mu, \pi_1), \mathbb{P}^t(\mu, \pi_2)). \tag{26}$$

Using Lemma A.2, we can take another step to relax the inequality (26) and write

$$W(\rho_\mu^{\pi_1}, \rho_\mu^{\pi_2}) \leq \frac{L_\pi(1 - \gamma)W(\pi_1, \pi_2)}{(L_\mu - 1)} \sum_{t=0}^{\infty} ((\gamma L_\mu)^t - \gamma^t). \tag{27}$$

Due to the $\gamma L_\mu < 1$ assumption, the right-hand summation in (27) is convergent. This leads us to

$$W(\rho_\mu^{\pi_1}, \rho_\mu^{\pi_2}) \leq \frac{L_\pi(1 - \gamma)W(\pi_1, \pi_2)}{(L_\mu - 1)} (\frac{1}{1 - \gamma L_\mu} - \frac{1}{1 - \gamma}). \tag{28}$$

Inequality (28) can then be simplified to give the result. $\qquad\square$

## A.4 STEPS TO BOUND THE SECOND-ORDER TERM

The RK duality yields the following bound:

$$|\langle \rho_\mu^{\pi_2} - \rho_\mu^{\pi_1}, A^{\pi_1}(\cdot, \pi_2) \rangle_s| \leq W(\rho_\mu^{\pi_1}, \rho_\mu^{\pi_2}) \cdot \sup_s \|\nabla_s A^{\pi_1}(s, \pi_2)\|_2. \tag{29}$$

To facilitate the further application of the RK duality, any advantage can be rewritten as the following inner product: $A^{\pi_1}(s, \pi_2) = \langle \pi_2(a|s) - \pi_1(a|s), Q^{\pi_1}(s, a) \rangle_a$. Taking derivatives of both sides with respect to the state variable and applying the triangle inequality produces the bound

$$\sup_s \|\nabla_s A^{\pi_1}(s, \pi_2)\|_2 \leq \sup_s \|\langle \nabla_s(\pi_2(a|s) - \pi_1(a|s)), Q^{\pi_1}(s, a) \rangle_a\|_2$$
$$+ \sup_s \|\langle \pi_2(a|s) - \pi_1(a|s), \nabla_s Q^{\pi_1}(s, a) \rangle_a\|_2. \tag{30}$$

The second term of the RHS in (30) is compatible with the RK duality. However, the form of the first term does not warrant an easy application of RK. For this, we introduce Theorem A.5.

**Theorem A.5.** *Assuming the existence of $\mathrm{Lip}(Q^{\pi_1}(s, a); a)$, we have the bound*

$$\left\| \langle \nabla_s(\pi_2(a|s) - \pi_1(a|s)), Q^{\pi_1}(s, a) \rangle_a \right\|_2 \tag{31}$$
$$\leq 2 \cdot \mathrm{Lip}(Q^{\pi_1}(s, a); a) \cdot \left\| \nabla_{s'} W\left( \frac{\pi_2(a|s') + \pi_1(a|s)}{2}, \frac{\pi_2(a|s) + \pi_1(a|bs')}{2} \right) \Big|_{s'=s} \right\|_2.$$

*Proof.* By definition, we have

$$\left\| \langle \nabla_s(\pi_2(a|s) - \pi_1(a|s)), Q^{\pi_1}(s, a) \rangle_a \right\|_2$$
$$= \sqrt{\sum_{j=1}^{\dim(\mathcal{S})} \left( \langle \frac{\partial}{\partial s^{(j)}}(\pi_2(a|s) - \pi_1(a|s)), Q^{\pi_1}(s, a) \rangle_a \right)^2}. \tag{32}$$

For better insight, we will write the derivative using finite differences:

$$\langle \frac{\partial}{\partial s^{(j)}}(\pi_2(a|s) - \pi_1(a|s)), Q^{\pi_1}(s, a) \rangle_a$$
$$= \lim_{\delta s \to 0} \frac{1}{\delta s} \Big[ \langle (\pi_2(a|s + \delta s \cdot \mathbf{e}_j) \qquad - \pi_1(a|s + \delta s \cdot \mathbf{e}_j) \qquad), Q^{\pi_1}(s, a) \rangle_a$$
$$- \langle (\pi_2(a|s) \qquad - \pi_1(a|s) \qquad ), Q^{\pi_1}(s, a) \rangle_a \Big]. \tag{33}$$

We can rearrange the finite difference terms to get

$$
\left\langle \frac{\partial}{\partial s^{(j)}}(\pi_2(a|s) - \pi_1(a|s)), Q^{\pi_1}(s,a) \right\rangle_a
$$
$$
= \lim_{\delta s \to 0} \frac{1}{\delta s} \Big[ \left\langle (\pi_2(a|s + \delta s \cdot \mathbf{e}_j) \quad\quad + \pi_1(a|s) \quad\quad\quad ), Q^{\pi_1}(s,a) \right\rangle_a
$$
$$
- \left\langle (\pi_2(a|s) \quad\quad\quad + \pi_1(a|s + \delta s \cdot \mathbf{e}_j) \quad ), Q^{\pi_1}(s,a) \right\rangle_a \Big]. \tag{34}
$$

Equivalently, we can divide and multiply the inner products by a factor of 2, to make the inner product arguments resemble mixture distributions:

$$
\left\langle \frac{\partial}{\partial s^{(j)}}(\pi_2(a|s) - \pi_1(a|s)), Q^{\pi_1}(s,a) \right\rangle_a
$$
$$
= \lim_{\delta s \to 0} \frac{2}{\delta s} \Big[ \left\langle \frac{\pi_2(a|s + \delta s \cdot \mathbf{e}_j) + \pi_1(a|s)}{2}, Q^{\pi_1}(s,a) \right\rangle_a
$$
$$
- \left\langle \frac{\pi_2(a|s) + \pi_1(a|s + \delta s \cdot \mathbf{e}_j)}{2}, Q^{\pi_1}(s,a) \right\rangle_a \Big]. \tag{35}
$$

The RK duality can now be used to bound this difference as

$$
\left| \left\langle \frac{\partial}{\partial s^{(j)}}(\pi_2(a|s) - \pi_1(a|s)), Q^{\pi_1}(s,a) \right\rangle_a \right| \tag{36}
$$
$$
\leq \lim_{\delta s \to 0} \frac{2}{\delta s} \Big[ W \left( \frac{\pi_2(a|s + \delta s \cdot \mathbf{e}_j) + \pi_1(a|s)}{2}, \frac{\pi_2(a|s) + \pi_1(a|s + \delta s \cdot \mathbf{e}_j)}{2} \right) \cdot \text{Lip}(Q^{\pi_1}(s,a); a) \Big],
$$

which can be simplified as

$$
\left| \left\langle \frac{\partial}{\partial s^{(j)}}(\pi_2(a|s) - \pi_1(a|s)), Q^{\pi_1}(s,a) \right\rangle_a \right|
$$
$$
\leq 2 \cdot \text{Lip}(Q^{\pi_1}(s,a); a) \cdot \frac{\partial}{\partial s'^{(j)}} W \left( \frac{\pi_2(a|s') + \pi_1(a|s)}{2}, \frac{\pi_2(a|s) + \pi_1(a|s')}{2} \right) \Big|_{s'=s}. \tag{37}
$$

Combining Inequality (37) and Equation (32), we obtain the bound in the theorem. $\qquad\square$

## A.5 THE LIPSCHITZ CONTINUITY AND TRANSITION STABILITY ASSUMPTIONS

There are three key groups of assumptions made in the derivation of our policy improvement lower bound. First is the existence of $Q^\pi$-function Lipschitz constants. Second is the transition dynamics Lipschitz-continuity assumptions. Finally, we make an assumption about the stability of the transition dynamics. Next, we will discuss the meaning and the necessity of these assumptions in the same order.

### A.5.1 ON THE EXISTENCE OF THE $\text{Lip}(Q^\pi, a)$ CONSTANT

The $\text{Lip}(Q^\pi, a)$ constant may be undefined when either the reward function or the transition dynamics are discontinuous. Examples of known environments with undefined $\text{Lip}(Q^\pi, a)$ constants include those with grazing contacts which define a discontinuous transition dynamics. In practice, even for environments that do not satisfy Lipschitz continuity assumptions, there are mitigating factors; practical $Q^\pi$ functions are reasonably narrow-bounded in a small trust region neighborhood, and since we use non-vanishing exploration scales and trust regions, a bounded interpolation slope can still model the $Q$-function variation effectively. We should also note that a slightly stronger version of this assumption is frequently used in the context of Lipschitz MDPs (Pirotta et al., 2015; Rachelson & Lagoudakis, 2010; Asadi et al., 2018). In practice, we have not found this to be a substantial limitation.

### A.5.2 THE TRANSITION DYNAMICS LIPSCHITZ CONTINUITY ASSUMPTION

Assumptions 5 and 6 of the main paper essentially represent the Lipschitz continuity assumptions of the transition dynamics with respect to actions and states, respectively. If the transition dynamics

and the policy are deterministic, then these assumptions are exactly equivalent to the Lipschitz continuity assumptions. Assumptions 5 and 6 only generalize the Lipschitz continuity assumptions in a distributional sense.

The necessity of these assumptions is a consequence of using metric measures for bounding errors. Traditional non-metric bounds force the use of full-support stochastic policies where all actions have non-zero probabilities (e.g., for the KL-divergence of two policies to be defined, TRPO needs to operate on full-support policies such as the Gaussian policies). In those analyses, since all policies share the same support, the next state distribution automatically becomes smooth and Lipschitz continuous with respect to the policy measure even if the transition dynamics was not originally smooth with respect to its input actions. However, metric measures are also defined for policies of non-overlapping support. To be able to provide closeness bounds for future state visitations of two similar policies with non-overlapping support, it becomes necessary to assume that close-enough actions or states must be yielding close-enough next states. In fact, this is a very common assumption in the framework of Lipschitz MDPs (See Section 2.2 of Rachelson & Lagoudakis (2010), Section 3 of Asadi et al. (2018), and Assumption 1 of Pirotta et al. (2015)).

### A.5.3 THE TRANSITION DYNAMICS STABILITY ASSUMPTION

Before moving to relax the $\gamma L_\mu < 1$ assumption, we will make a few definitions and restate the previous lemmas and theorems under these definitions. We define $L_{\mu_1,\mu_2,\pi}$ to be the infimum non-negative value that makes the equation $W(\mathbb{P}(\mu_1,\pi),\mathbb{P}(\mu_2,\pi)) = L_{\mu_1,\mu_2,\pi}W(\mu_1,\mu_2)$ hold. Similarly, $L_{\mu_1,\mu_2,\pi}$ is defined as the infimum non-negative value that makes the equation $W(\mathbb{P}(\mu,\pi_1),\mathbb{P}(\mu,\pi)) = L_{\mu,\pi_1,\pi_2}W(\pi_1(\cdot|\mu),\pi_2(\cdot|\mu))$ hold. For notation brevity, we will also denote $L_{\mathbb{P}^t(\mu,\pi_1),\mathbb{P}^t(\mu,\pi_2),\pi_2}$ and $L_{\mathbb{P}^t(\mu,\pi_1),\pi_1,\pi_2}$ by $\tilde{L}^{(t)}_{\mu,\pi_1,\pi_2}$ and $\hat{L}^{(t)}_{\mu,\pi_1,\pi_2}$, respectively.

Under these definitions, Lemma A.1 evolves into

$$W(\mathbb{P}(\mu_1,\pi_1),\mathbb{P}(\mu_2,\pi_2)) \le L_{\mu_1,\mu_2,\pi}W(\mu_1,\mu_2) + L_{\mu_1,\pi_1,\pi_2}W(\pi_1,\pi_2). \tag{38}$$

We can apply a time-point recursion to this lemma and have

$$W(\mathbb{P}(\mathbb{P}^t(\mu,\pi_1),\pi_1),\mathbb{P}(\mathbb{P}^t(\mu,\pi_2),\pi_2))$$
$$\le L_{\mathbb{P}^t(\mu,\pi_1),\pi_1,\pi_2}W(\pi_1,\pi_2) + L_{\mathbb{P}^t(\mu,\pi_1),\mathbb{P}^t(\mu,\pi_2),\pi_2}W(\mathbb{P}^t(\mu,\pi_1),\mathbb{P}^t(\mu,\pi_2)) \tag{39}$$

, which can be notationally simplified to

$$W(\mathbb{P}^t(\mu,\pi_1),\mathbb{P}^t(\mu,\pi_2)) \le \hat{L}^{(t-1)}_{\mu,\pi_1,\pi_2}W(\pi_1,\pi_2) + \tilde{L}^{(t-1)}_{\mu,\pi_1,\pi_2}W(\mathbb{P}^{t-1}(\mu,\pi_1),\mathbb{P}^{t-1}(\mu,\pi_2)). \tag{40}$$

These modifications lead Lemma A.2 to be updated accordingly into

$$W(\mathbb{P}^t(\mu,\pi_1),\mathbb{P}^t(\mu,\pi_2)) \le C^{(t)}_{L;\mu,\pi_1,\pi_2} \cdot W(\pi_1,\pi_2) \tag{41}$$

, where we have

$$C^{(t)}_{L;\mu,\pi_1,\pi_2} := \sum_{k=1}^{t} \hat{L}^{(t-k)}_{\mu,\pi_1,\pi_2} \prod_{i=1}^{k-1} \tilde{L}^{(t-i)}_{\mu,\pi_1,\pi_2}. \tag{42}$$

By a simple change of variables, we can have the equivalent definition of

$$C^{(t)}_{L;\mu,\pi_1,\pi_2} := \sum_{k=1}^{t} \hat{L}^{(k-1)}_{\mu,\pi_1,\pi_2} \prod_{i=k+1}^{t-1} \tilde{L}^{(i)}_{\mu,\pi_1,\pi_2}. \tag{43}$$

Now, we would replace the $\gamma L_\mu < 1$ assumption with the following assumption.

**The Transition Dynamics Stability Assumption**: A transition dynamics $P$ is called stable if and only if the induced $\{\tilde{L}^{(t)}_{\mu,\pi_1,\pi_2}\}_{t\ge0}$ and $\{\hat{L}^{(t)}_{\mu,\pi_1,\pi_2}\}_{t\ge0}$ sequences satisfy

$$C_L := \sup_{\mu,\pi_1,\pi_2,t} C^{(t)}_{L;\mu,\pi_1,\pi_2} = \sup_{\mu,\pi_1,\pi_2,t} \sum_{k=1}^{t} \hat{L}^{(k-1)}_{\mu,\pi_1,\pi_2} \prod_{i=k+1}^{t-1} \tilde{L}^{(i)}_{\mu,\pi_1,\pi_2} < \infty. \tag{44}$$

To help understand which $\{\tilde{L}^{(t)}_{\mu,\pi_1,\pi_2}\}_{t\ge0}$ and $\{\hat{L}^{(t)}_{\mu,\pi_1,\pi_2}\}_{t\ge0}$ sequences can satisfy this assumption, we will provide some examples:

- The $\forall t : \tilde{L}^{(t)}_{\mu,\pi_1,\pi_2} = c_1 > 1$ and $\forall t : \hat{L}^{(t)}_{\mu,\pi_1,\pi_2} = c_2$ sequences violate the dynamics stability assumption.

- The $\forall t : \tilde{L}^{(t)}_{\mu,\pi_1,\pi_2} \leq 1$ and $\forall t : \hat{L}^{(t)}_{\mu,\pi_1,\pi_2} = O(\frac{1}{t^2})$ sequences satisfy the dynamics stability assumption.

- $\sup_t \tilde{L}^{(t)}_{\mu,\pi_1,\pi_2} < 1$ guarantees the dynamics stability assumption.

- $\forall t \geq t_0 : \tilde{L}^{(t)}_{\mu,\pi_1,\pi_2} < 1$ guarantees the dynamics stability assumption no matter (1) how big $t_0$ is (as long as it is finite), or (2) how big the members of the finite set $\{\tilde{L}^{(t)}_{\mu,\pi_1,\pi_2} | t < t_0\}$ are.

If the dynamics stability assumption holds with a constant $C_L$, one can define a $\bar{L}_\mu$ constant such that $C_L = L_\pi \sum_{t=0}^{\infty} (\gamma \bar{L}_\mu)^t$. Then, we can replace all the $L_\mu$ instances in the rest of the proof with the corresponding $\bar{L}_\mu$ constant, and the results will remain the same without any change of format.

The $\tilde{L}^{(t)}_{\mu,\pi_1,\pi_2}$ and $\hat{L}^{(t)}_{\mu,\pi_1,\pi_2}$ constants can be thought as tighter versions of $L_\mu$ and $L_\pi$, but with dependency on $\pi_1$, $\pi_2$, $\mu$ and the time-point of application. Having $\gamma L_\mu < 1$ is a sufficient yet unnecessary condition for this dynamics stability assumption to hold. Vaguely speaking, $L_\mu$ is an expansion rate for the state distribution distance; it tells you how much a divergence in the state distribution will expand after a single application of the transition dynamics. Having effective expansion rates that are larger than one throughout an infinite horizon trajectory is a sign of the system instability; some change in the initial state's distribution could cause the observations to diverge exponentially. While controlling unstable systems is an important and practical challenge, none of the existing reinforcement learning methods is capable of learning effective policies on such environments. Roughly speaking, having the dynamics stability assumption guarantees that the expansion rates cannot be consistently larger than one for infinite time steps.

## A.6  CHOICE OF $C_1$ AND $C_2$

Since the TDPO algorithm operates using the metric Wasserstein distance, thinking about how normalizing actions and rewards affect the corresponding optimization objective builds insight into how to set these coefficients properly. Say we use the same dynamics, only the new actions are scaled up by a factor of $\beta$ and the rewards are scaled up by a factor of $\alpha$:

$$a_{\text{new}} = \beta \cdot a_{\text{old}} \qquad r_{\text{new}} = \alpha \cdot r_{\text{old}}. \tag{45}$$

If the policy function approximation class remained the same, the policy gradient would be scaled by a factor of $\frac{\alpha}{\beta}$ (i.e., $\frac{\partial \eta_{\text{new}}}{\partial a_{\text{new}}} = \frac{\alpha}{\beta} \cdot \frac{\partial \eta_{\text{old}}}{\partial a_{\text{old}}}$). Therefore, one can easily show that the corresponding new regularization coefficient and trust region sizes can be obtained by

$$C_{\text{new}} = \frac{\alpha}{\beta^2} \cdot C_{\text{old}} \tag{46}$$

and

$$\delta_{\text{max}}^{\text{new}} = \beta \cdot \delta_{\text{max}}^{\text{old}}. \tag{47}$$

We used equal regularization coefficients (i.e., $C_1 = C_2 = C$), and the process to choose them can be summarized as follows: (1) Define $C = 3600 \cdot \alpha \cdot \beta^{-2}$, $\delta_{\text{max}} = \beta/600$ and $\sigma_q = \beta/60$ (where $\sigma_q$ is the action disturbance parameter used for DeVine), (2) using prior knowledge or by trial and error determine appropriate action and reward normalization coefficients. The reward normalization coefficient $\alpha$ was sought to be approximately the average per-step discounted reward difference of a null policy and an optimal policy. We used a reward scaling value of $\alpha = 5$ and an action scaling value of $\beta = 5$ for the non-locally rewarded pendulum and $\beta = 1.5$ for the long-horizon legged robot. Both environments had a per-step discounted reward of approximately $-5$ for a null policy and non-positive rewards, justifying the choice of $\alpha$.

## A.7  PROOF OF THEOREM 4.1

We restate Theorem 4.1 below for reference and now prove it.

**Theorem 4.1.** *Assume a finite horizon MDP with both deterministic transition dynamics $P$ and initial distribution $\mu$, with maximal horizon length of $H$. Define $K = H \cdot \dim(\mathcal{A})$, a uniform $\nu$, and $q(s; \sigma) = \pi_1(s) + \sigma \mathbf{e}_j$ in Algorithm 2 with $\mathbf{e}_j$ being the $j^{th}$ basis element for $\mathcal{A}$. If the $(j, t_k)$ pairs are sampled to exactly cover $\{1, \ldots, \dim(\mathcal{A})\} \times \{1, \ldots, H\}$, then we have*

$$\lim_{\sigma \to 0} \nabla_{\pi_2} \mathbb{A}^{\pi_1}(\pi_2)\big|_{\pi_2 = \pi_1} = \nabla_{\pi_2} \eta_{\pi_2}\big|_{\pi_2 = \pi_1}. \tag{48}$$

*Proof.* According to the advantage decomposition lemma, we have

$$\nabla_{\pi_2} \eta_{\pi_2}\big|_{\pi_2 = \pi_1} = \frac{1}{1 - \gamma} \mathbb{E}_{s \sim \rho_\mu^{\pi_1}} \left[ \nabla_{\pi_2} A^{\pi_1}(s, \pi_2) \right] \big|_{\pi_2 = \pi_1}. \tag{49}$$

Due to the fact that the transition dynamics, policies $\pi_1$ and $\pi_2$, and initial state distribution are all deterministic, we can simplify Equation (49) to

$$\nabla_{\pi_2} \eta_{\pi_2}\big|_{\pi_2 = \pi_1} = \sum_{t=0}^{H-1} \gamma^t \cdot \nabla_{\pi_2} A^{\pi_1}(s_t, \pi_2)\big|_{\pi_2 = \pi_1}, \tag{50}$$

where $s_t$ is the state after applying the policy $\pi_1$ for $t$ time-steps. We can use the chain rule to write

$$\nabla_{\pi_2} A^{\pi_1}(s_t, \pi_2)\big|_{\pi_2 = \pi_1} = \nabla_{\pi_2} A^{\pi_1}(s_t, a_t)\big|_{\substack{a_t = \pi_2(s_t) \\ \pi_2 = \pi_1}}$$

$$= \sum_{j=1}^{\dim(\mathcal{A})} \nabla_{\pi_2} a_t^{(j)} \cdot \frac{\partial}{\partial a_t^{(j)}} A^{\pi_1}(s_t, a_t)\big|_{\substack{a_t = \pi_2(s_t) \\ \pi_2 = \pi_1}}. \tag{51}$$

To recap, Equations (50), (50), and (51) can be summarized as

$$\nabla_{\pi_2} \eta_{\pi_2}\big|_{\pi_2 = \pi_1} = \sum_{t=0}^{H-1} \gamma^t \sum_{j=1}^{\dim(\mathcal{A})} \nabla_{\pi_2} a_t^{(j)} \cdot \frac{\partial}{\partial a_t^{(j)}} A^{\pi_1}(s_t, a_t)\big|_{\substack{a_t = \pi_2(s_t) \\ \pi_2 = \pi_1}}. \tag{52}$$

Under the assumption that the $(j, t)$ pairs are sampled to exactly cover $\{1, \ldots, \dim(\mathcal{A})\} \times \{1, \ldots, H\}$, we can simplify the DeVine oracle to

$$\mathbb{A}^{\pi_1}(\pi_2) = \frac{1}{K} \sum_{t=0}^{H-1} \sum_{j=1}^{\dim(\mathcal{A})} \left[ \frac{\dim(\mathcal{A}) \cdot \gamma^t}{\nu(t)} \cdot \frac{(\pi_2(s_t) - \pi_1(s_t))^T (q(s_t; j, \sigma) - \pi_1(s_t))}{(q(s_t; j, \sigma) - \pi_1(s_t))^T (q(s_t; j, \sigma) - \pi_1(s_t))} \right.$$

$$\left. \cdot A^{\pi_1}(s_t, q(s_t; j, \sigma)) \right]. \tag{53}$$

From the $q$ definition, we have $q(s_t; j, \sigma) - \pi_1(s_t) = \sigma \mathbf{e}_j$ and $(q(s_t; j, \sigma) - \pi_1(s_t))^T (q(s_t; j, \sigma) - \pi_1(s_t)) = \sigma^2$. Since $\nu$ is uniform (i.e., $\nu(t) = \frac{1}{H}$) and $K = H \cdot \dim(\mathcal{A})$, we can take the policy gradient of Equation (53) and simplify it into

$$\nabla_{\pi_2} \mathbb{A}^{\pi_1}(\pi_2)\big|_{\pi_2 = \pi_1} = \sum_{t=0}^{H-1} \sum_{j=1}^{\dim(\mathcal{A})} \left[ \gamma^t \cdot \nabla_{\pi_2} (\pi_2(s_t) - \pi_1(s_t))^T \mathbf{e}_j \cdot \frac{A^{\pi_1}(s_t, q(s_t; j, \sigma))}{\sigma} \right]. \tag{54}$$

Since, $A^{\pi_1}(s_t, \pi_1(s_t)) = 0$, we can write

$$\lim_{\sigma \to 0} \frac{A^{\pi_1}(s_t, q(s_t; j, \sigma))}{\sigma} = \lim_{\sigma \to 0} \frac{A^{\pi_1}(s_t, \pi_1(s_t) + \sigma \mathbf{e}_j) - A^{\pi_1}(s_t, \pi_1(s_t))}{\sigma}$$

$$= \frac{\partial}{\partial a_t^{(j)}} A^{\pi_1}(s_t, a_t)\big|_{a_t = \pi_1(s_t)}. \tag{55}$$

Also, by the definition of the gradient, we can write

$$\nabla_{\pi_2} (\pi_2(s_t) - \pi_1(s_t))^T \mathbf{e}_j = \nabla_{\pi_2} a_t^{(j)}. \tag{56}$$

Combining Equations (55) and (56), and applying them to Equation (54), yields

$$\lim_{\sigma \to 0} \nabla_{\pi_2} \mathbb{A}^{\pi_1}(\pi_2)\big|_{\pi_2 = \pi_1} = \sum_{t=0}^{H-1} \sum_{j=1}^{\dim(\mathcal{A})} \gamma^t \cdot \nabla_{\pi_2} a_t^{(j)} \cdot \frac{\partial}{\partial a_t^{(j)}} A^{\pi_1}(s_t, a_t)\big|_{\substack{a_t = \pi_2(s_t) \\ \pi_2 = \pi_1}}. \tag{57}$$

Finally, the theorem can be obtained by comparing Equations (57) and (52).

$$\square$$

## A.8 QUADRATIC MODELING OF POLICY SENSITIVITY REGULARIZATION

First, we will build insight into the nature of the

$$\mathcal{L}_{WG}(\pi_1, \pi_2; s) = W(\pi_2(a|s), \pi_1(a|s)) \times$$

$$\left\| \nabla_{s'} W \left( \frac{\pi_2(a|s') + \pi_1(a|s)}{2}, \frac{\pi_2(a|s) + \pi_1(a|s')}{2} \right) \Big|_{s'=s} \right\|_2 \tag{58}$$

term. It is fairly obvious that

$$W(\pi_2(a|s), \pi_1(a|s)) \big|_{\pi_2=\pi_1} = 0. \tag{59}$$

If $\pi_2 = \pi_1$, then the two distributions $\frac{\pi_2(a|s') + \pi_1(a|s)}{2}$ and $\frac{\pi_2(a|s) + \pi_1(a|s')}{2}$ will be the same no matter what $s'$ is. In other words,

$$\pi_1 = \pi_2 \Rightarrow \forall s' : W \left( \frac{\pi_2(a|s') + \pi_1(a|s)}{2}, \frac{\pi_2(a|s) + \pi_1(a|s')}{2} \right) = 0. \tag{60}$$

This means that

$$\left\| \nabla_{s'} W \left( \frac{\pi_2(a|s') + \pi_1(a|s)}{2}, \frac{\pi_2(a|s) + \pi_1(a|s')}{2} \right) \Big|_{s'=s} \right\|_2 \Big|_{\pi_2=\pi_1} = 0. \tag{61}$$

The Taylor expansion of the squared Wasserestein distance can be written as

$$W(\pi_2(a|s), \pi_1(a|s))^2 \big|_{\theta_2=\theta_1+\delta\theta} = \frac{1}{2} \delta\theta^T H_2 \delta\theta + \text{h.o.t.}. \tag{62}$$

Considering (60) and similar to the previous point, one can write the following Taylor expansion

$$\left\| \nabla_{s'} W \left( \frac{\pi_2(a|s') + \pi_1(a|s)}{2}, \frac{\pi_2(a|s) + \pi_1(a|s')}{2} \right) \Big|_{s'=s} \right\|_2^2 \Big|_{\theta_2=\theta_1+\delta\theta} = \delta\theta^T H_1 \delta\theta + \text{h.o.t.}. \tag{63}$$

According to above, $\mathcal{L}_{WG}$ is the geometric mean of two functions of quadratic order. Although this makes $\mathcal{L}_{WG}$ of quadratic order (i.e., $\lim_{\delta\theta \to 0} \frac{\mathcal{L}_{WG}(\alpha\delta\theta)}{\mathcal{L}_{WG}(\delta\theta)} = \alpha^2$ holds for any constant $\alpha$), this does not guarantee that $\mathcal{L}_{WG}$ is twice continuously differentiable w.r.t. the policy parameters, and may not have a defined Hessian matrix (e.g., $f(x_1, x_2) = |x_1 x_2|$ is of quadratic order, yet is not twice differentiable). To avoid this issue, we compromise on the local model. Using the AM-GM inequality and for any arbitrary positive $\alpha$, one can bound the $\mathcal{L}_{WG}$ term into two quadratic terms:

$$\mathcal{L}_{WG}(\pi_1, \pi_2; s) \leq \frac{1}{2} \left( \frac{1}{\alpha^2} W(\pi_2(a|s), \pi_1(a|s))^2 + \right.$$

$$\left. \alpha^2 \left\| \nabla_{s'} W \left( \frac{\pi_2(a|s') + \pi_1(a|s)}{2}, \frac{\pi_2(a|s) + \pi_1(a|s')}{2} \right) \Big|_{s'=s} \right\|_2^2 \right). \tag{64}$$

Therefore, by defining

$$\mathcal{L}_{G^2}(\pi_1, \pi_2; s) := \left\| \nabla_{s'} W \left( \frac{\pi_2(a|s') + \pi_1(a|s)}{2}, \frac{\pi_2(a|s) + \pi_1(a|s')}{2} \right) \Big|_{s'=s} \right\|_2^2, \tag{65}$$

$C_1' := \frac{C_1 \cdot \alpha^2}{2}$, and $C_2' := (C_2 + \frac{C_1}{2\alpha^2})$ the new surrogate will have the twice-differentiable form

$$\mathcal{L}_{\pi_1}(\pi_2) = \frac{1}{1-\gamma} \cdot \mathbb{E}_{s \sim \rho_\mu^{\pi_1}} [A^{\pi_1}(s, \pi_2)]$$

$$- C_1' \cdot \mathbb{E}_{s \sim \rho_\mu^{\pi_1}} \left[ \mathcal{L}_{G^2}(\pi_1, \pi_2; s) \right]$$

$$- C_2' \cdot \mathbb{E}_{s \sim \rho_\mu^{\pi_1}} \left[ W(\pi_2(a|s), \pi_1(a|s))^2 \right]. \tag{66}$$

$C_1'$ and $C_2'$ will be the corresponding regularization coefficients for the surrogate defined in (13). Due to the arbitrary $\alpha$ used in bounding, no constrain governs the $C_1'$ and $C_2'$ coefficients. Therefore, $C_1'$ and $C_2'$ can be chosen without constraining each other.

A.9  IMPLEMENTATION DETAILS FOR THE ENVIRONMENT WITH NON-LOCAL REWARDS

We used the stable-baselines implementation (Hill et al., 2018), which has the same structure as the original OpenAI baselines (Dhariwal et al., 2017) implementation. We used the "ppo1" variant since no hardware acceleration was necessary for automatic differentiation and MPI parallelization was practically efficient. TDPO, TRPO, and PPO used the same function approximation architecture with two hidden layers, 64 units in each layer, and the tanh activation. TRPO, PPO, DDPG, and TD3 used their default hyper-parameter settings. We used Xavier initialization (Glorot & Bengio, 2010) for TDPO, and multiplied the outputs of the network by a factor of 0.001 so that the initial actions were small and nearly zero. We also confirmed that TDPO does not decrease in performance when using an identical network initialization to PPO/TRPO. TD3's baseline implementation was amended to support MPI parallelization just like TRPO, PPO, and DDPG. To produce the results for DDPG and TD3, we used hyperparameter optimization both with and without the tanh final activation function that is common for DDPG and TD3 (this causes the difference in initial payoff in the figures). However, under no conditions were DDPG and TD3 able to solve these environments effectively, suggesting that the deterministic search used by TDPO is operating in a qualitatively different way than the stochastic policy optimization used by DDPG and TD3. Note that we made a thorough attempt to compare DDPG and TD3 fairly, including trying different initializations, different final layer scalings/activations, different network architectures, and performing hyperparameter optimization. Mini-batch selection was unnecessary for TDPO since optimization for samples generated by DeVine was fully tractable. The confidence intervals in all figures were generated using 1000 samples of the statistics of interest.

For designing the environment, we used Dhariwal et al. (2017)'s pendulum dynamics and relaxed the torque thresholds to be as large as $40$ N m. The environment also had the same episode length of 200 time-steps. We used the reward function described by the following equations:

$$
\begin{aligned}
R(s_t, a_t) &= C_R \cdot R(\tau) \cdot \mathbf{1}\{t = 200\} \\
R(\tau) &= R_{\text{Freq}}(\tau) + R_{\text{Offset}}(\tau) + R_{\text{Amp}}(\tau) \\
R_{\text{Freq}}(\tau) &= 0.1 \cdot \left[ \sum_{f=f_{\min}}^{f_{\max}} \Theta_{\text{std}}^+(f)^2 - 1 \right] \\
R_{\text{Offset}}(\tau) &= -\left| \frac{\Theta(f = 0)}{200} - \theta_{\text{Target Offset}} \right| = -\left| \left( \frac{1}{200} \sum_{t=0}^{199} \theta_t \right) - \theta_{\text{Target Offset}} \right| \\
R_{\text{Amp}}(\tau) &= h_{\text{piecewise}} \left( \frac{\Theta_{\text{AC}}}{\theta_{\text{Target Amp.}}} - 1 \right)
\end{aligned}
$$

$$(67)$$

where

- $\theta$ is the pendulum angle signal in the time domain.
- $\Theta$ is the magnitude of the Fourier transform of $\theta$.
- $\Theta^+$ is the same as $\Theta$ only for the positive frequency components.
- $\Theta_{\text{AC}}$ is the normalized oscillatory spectrum of $\Theta$:

$$
\Theta_{\text{AC}} = \frac{\sqrt{\Theta^{+T} \Theta^+}}{200}. \tag{68}
$$

- $h_{\text{piecewise}}$ is a piece-wise linear error penalization function:

$$
h_{\text{piecewise}}(x) = -x \cdot \mathbf{1}\{x \geq 0\} + 10^{-4} x \cdot \mathbf{1}\{-x \geq 0\}. \tag{69}
$$

- $\Theta_{\text{std}}^+$ is the standardized positive amplitudes vector:

$$
\Theta_{\text{std}}^+ = \frac{\Theta^+}{\sqrt{\Theta^{+T} \Theta^+ + 10^{-6}}}. \tag{70}
$$

- $C_R = 1.3 \times 10^4$ is a reward normalization coefficient, and was chosen to yield approximately the same payoff as a null policy would yield in the typical pendulum environment of Dhariwal et al. (2017).

- $\theta_{\text{Target Offset}}$ is the target offset, $\theta_{\text{Target Amp.}}$ is the target amplitude, and $[f_{\min}, f_{\max}]$ is the target frequency range of the environment.

All methods used 48 parallel workers. The machines used Xeon E5-2690-v3 processors and 256 GB of memory. Each experiment was repeated 25 times for each method, and each run was given 6 hours or 500 million samples to finish.

## A.10 Implementation Details for the Environment with Long Horizon and Resonant Frequencies

For the robotic leg, we used exactly the same algorithms with the same parameters as described in Section A.9 above.

We used the reward function described by the following equations:

$$R = R_{\text{posture}} + R_{\text{velocity}} + R_{\text{foot offset}} + R_{\text{foot height}} + R_{\text{ground force}} + R_{\text{knee height}} + R_{\text{on-air torques}} \quad (71)$$

with

$$
\begin{aligned}
R_{\text{posture}} &= -1 \times \left[ \left| \theta_{\text{knee}} + \frac{\pi}{2} \right| + \left| \theta_{\text{hip}} + \frac{\pi}{4} \right| \right] \\
R_{\text{velocity}} &= -0.08 \times \left[ |\omega_{\text{knee}}| + |\omega_{\text{hip}}| \right] \\
R_{\text{foot offset}} &= -10 \times \left[ |x_{\text{foot}}| \cdot \mathbf{1}\{z_{\text{knee}} < 0.2\} \right] \\
R_{\text{ground force}} &= -1 \times \left[ |f_z - mg| \cdot \mathbf{1}\{f_z < mg\} \cdot \mathbf{1}_{\text{touchdown}} \right] \\
R_{\text{foot height}} &= -1 \times \left[ |z_{\text{foot}}| \cdot \mathbf{1}_{\text{touchdown}} \right] \\
R_{\text{knee height}} &= -15 \times \left[ \left| z_{\text{knee}} - z_{\text{knee}}^{\text{target}} \right| \cdot \mathbf{1}_{\text{touchdown}} \right] \\
R_{\text{on-air torques}} &= -10^{-4} \times \left[ (\tau_{\text{knee}}^2 + \tau_{\text{hip}}^2) \cdot (1 - \mathbf{1}_{\text{touchdown}}) \right]
\end{aligned}
\quad (72)
$$

where

- $\theta_{\text{knee}}$ and $\theta_{\text{hip}}$ are the knee and hip angles in radians, respectively.

- $\omega_{\text{knee}}$ and $\omega_{\text{hip}}$ are the knee and hip angular velocities in radians per second, respectively.

- $x_{\text{foot}}$ and $z_{\text{foot}}$ are the horizontal and vertical foot offsets in meters from the desired standing point on the ground, respectively.

- $x_{\text{knee}}$ and $z_{\text{knee}}$ are the horizontal and vertical knee offsets in meters from the desired standing point on the ground, respectively.

- $f_z$ is the vertical ground reaction force on the robot in Newtons.

- $m$ is the robot weight in kilograms (i.e., $m = 0.76$ kg).

- $g$ is the gravitational acceleration in meters per second squared.

- $\mathbf{1}_{\text{touchdown}}$ is the indicator function of whether the robot has ever touched the ground.

- $z_{\text{knee}}^{\text{target}}$ is a target knee height of 0.1 m.

- $\tau_{\text{knee}}$ and $\tau_{\text{hip}}$ are the knee and hip torques in Newton meters, respectively.

All methods used 72 full trajectories between each policy update, and each run was given 16 hours of wall time, which corresponded to almost 500 million samples. This experiment was repeated 75 times for each method. The empirical mean of the discounted payoff values were reported without any performance or seed filtration. The same hardware as the non-local rewards experiments (i.e., Xeon E5-2690-v3 processors and 256 GB of memory) was used.

## A.11 GYM SUITE BENCHMARKS

While it is clear that our deterministic policy gradient performs well on the new control environments we consider, one may naturally wonder about its performance on existing RL control benchmarks. We ran our method on a suite of Gym environments and include four representative examples in Figure 3. Broadly speaking, our method (TDPO) performs slightly worse on average than others, but occasionally performs much better as seen in the Swimmer-v3 environment. We speculate that these gym environments are reasonably robust to injected noise, and this may mean that stochastic policy gradients can more rapidly and efficiently explore the policy space, or there may be other algorithmic enhancements that are needed for fully deterministic policy gradients in these cases. The experiments granted each method 72 parallel MPI workers for about 144 million steps (i.e., 2 million sequential steps), and the returns were averaged over 100 different seeds for each method. Since the computational cost of running both DDPG and TD3 were high, we only included TD3 since it was shown to outperform DDPG in earlier benchmarks.

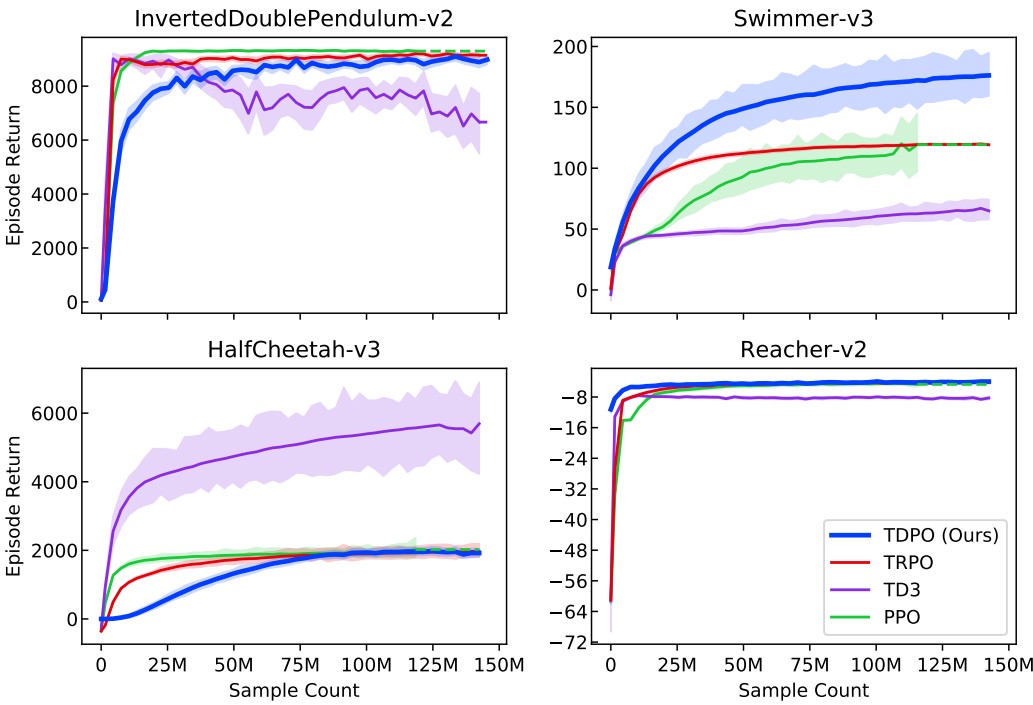

Figure 3: Results for the gym suite benchmarks.

## A.12 RUNNING TIME COMPARISON

Figure 4 depicts a comparison of each method's running time per million steps. These plots show the combination of both the simulation (i.e., environment sampling) and the optimization (i.e., computing the policy gradient and running the conjugate gradient solver) time. It is clear that our method (TDPO) is generally faster than the other algorithms. This is mainly due to the computational efficiency of the DeVine gradient estimator, which summarizes two full trajectories in a single state-action-advantage tuple which can significantly reduce the optimization time. That being said, these relative comparisons could vary to a large extent (1) under different processor architectures, (2) with more (or less) efficient implementations, or (3) when running environments whose simulation time constitutes a significantly larger (or smaller) portion of the total running time.

## A.13 OTHER SWINGING PENDULUM VARIANTS

Multiple variants of the pendulum with non-local rewards were used, each with different frequency targets and the same reward structure. Table 3 summarizes the target characteristics of each variant.

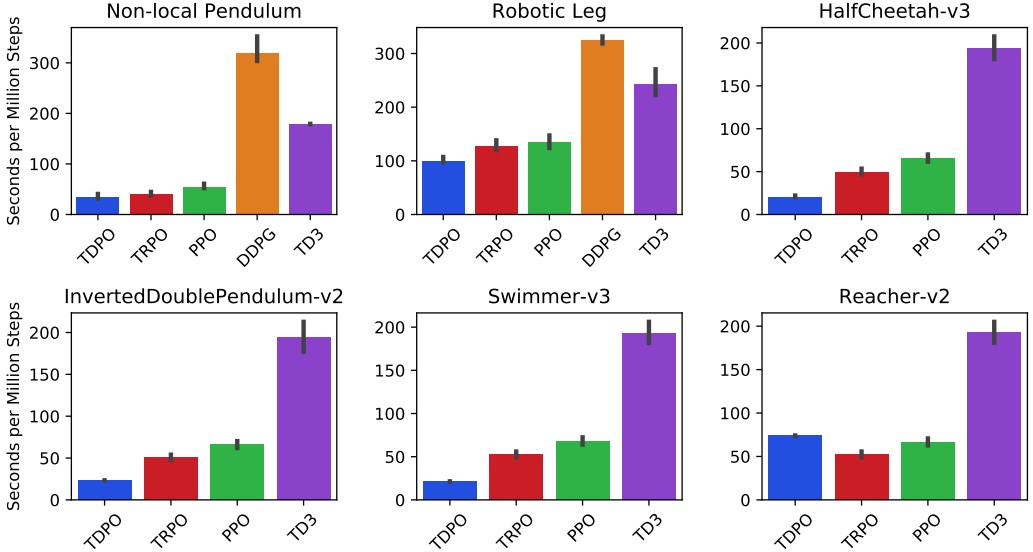

Figure 4: Training time comparison in different environments. The lower the bar, the faster the method. The vertical axis shows the time in seconds needed to consume one million state-action pairs for training. Each environment was shown separately in a different subplot.

The main variant was shown in the paper. Figures 5, 6, 7, 8, 9, 10, 11, and 12 show similar results for the second to ninth variants. To focus on our method's ability to solve all these variants efficiently, we only show the performance of our method (TDPO) on all variants in Figure 13. Overall, we found TRPO, PPO, DDPG, and TD3 to occasionally find the correct offset. They either excited the natural or the maximum (not the desired) frequency of the pendulum, but they were not able to drive the desired frequency and amplitude. TDPO was able to achieve the desired oscillations (and thus high rewards) in all variants.

| Pendulum Variant | Desired Frequency | Desired Offset | Desired Amplitude |
|---|---|---|---|
| Main | 1.7–2 Hz | 0.524 rad | 0.28 rad |
| Second | 0.5.–0.7 Hz | 1.571 rad | 1.11 rad |
| Third | 2.5–3 Hz | 0.524 rad | 0.28 rad |
| Fourth | 2.–2.4 Hz | 0.785 rad | 0.28 rad |
| Fifth | 2.–2.4 Hz | 1.571 rad | 0.74 rad |
| Sixth | 2–2.4 Hz | 0.524 rad | 0.28 rad |
| Seventh | 2.–2.4 Hz | 1.047 rad | 0.28 rad |
| Eighth | 2–2.4 Hz | 0.785 rad | 0.74 rad |
| Ninth | 2.–2.4 Hz | 1.309 rad | 0.28 rad |

Table 3: The target oscillation characteristics used to define different pendulum swinging environments.

## A.14 NOTES ON HOW TO IMPLEMENT TDPO

In short, our method (TDPO) is structured in the same way TRPO was structured; both TDPO and TRPO use policy gradient estimation, and a conjugate-gradient solver utilizing a Hessian-vector product machinery. On the other hand, there are some algorithmic differences that distinguish TDPO from TRPO. First of all, TRPO uses line-search heuristics to adaptively find the update scale; no such heuristics are applied in TDPO. Second, TDPO uses the DeVine advantage estimator, which requires storing and reloading pseudo-random generator states. Finally, the Hessian-vector product machinery used in TDPO computes Wasserstein-vector products, which is slightly different from those used in TRPO. The hyper-parameter settings and notes on how to choose them were discussed

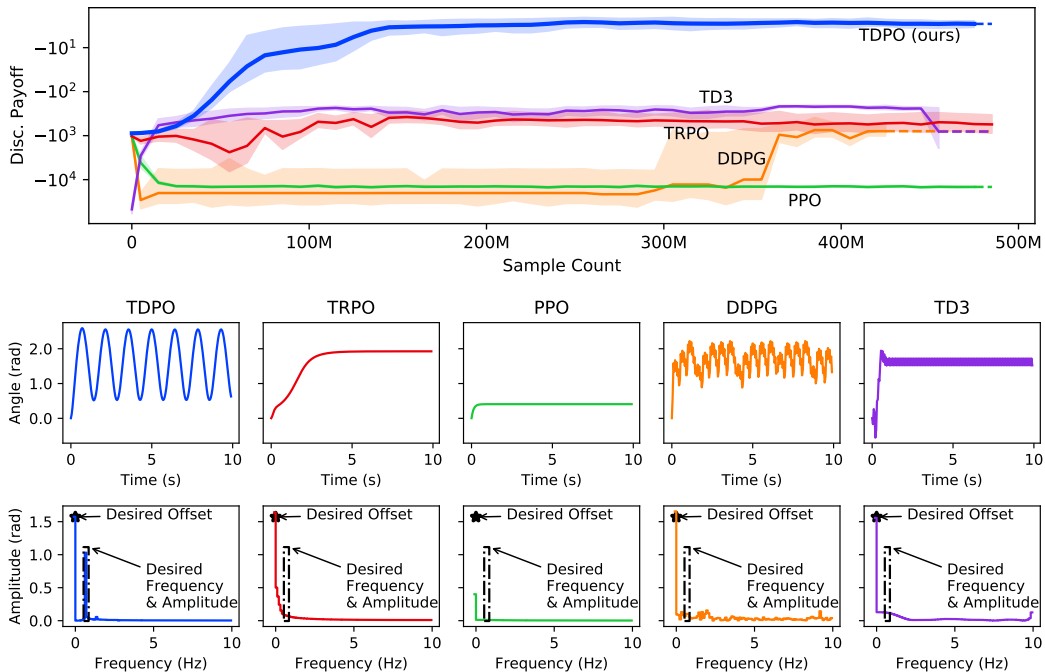

Figure 5: Results for the second variant of the simple pendulum with non-local rewards. Upper panel: training curves with empirical discounted payoffs. Lower panels: trajectories in both the time domain and frequency domain, showing target values of oscillation frequency, amplitude, and offset.

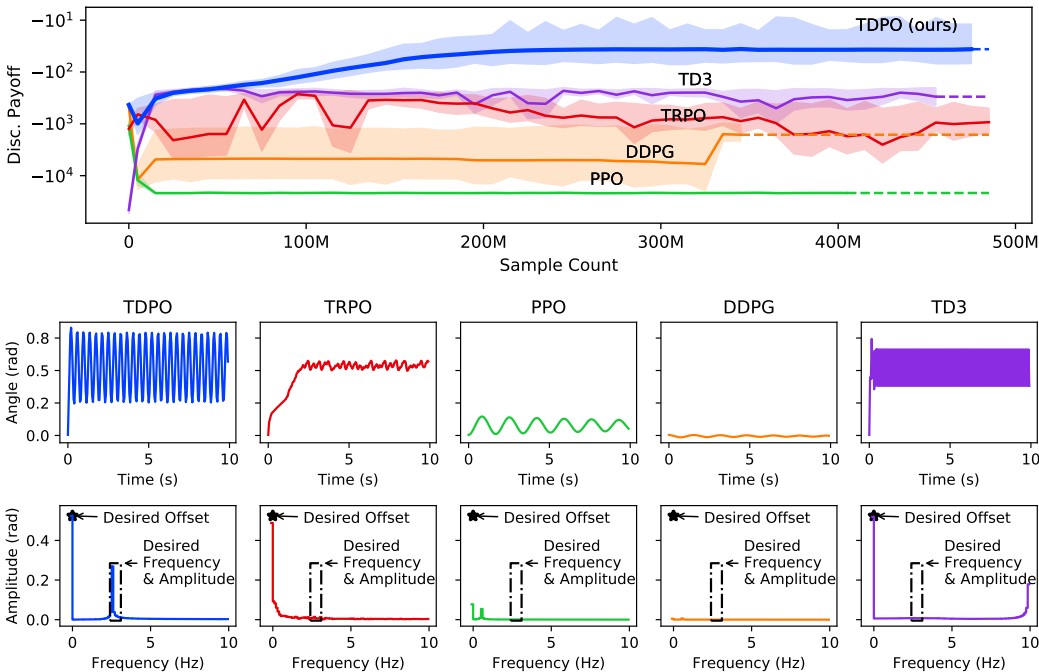

Figure 6: Results for the third variant of the simple pendulum with non-local rewards. Upper panel: training curves with empirical discounted payoffs. Lower panels: trajectories in both the time domain and frequency domain, showing target values of oscillation frequency, amplitude, and offset.

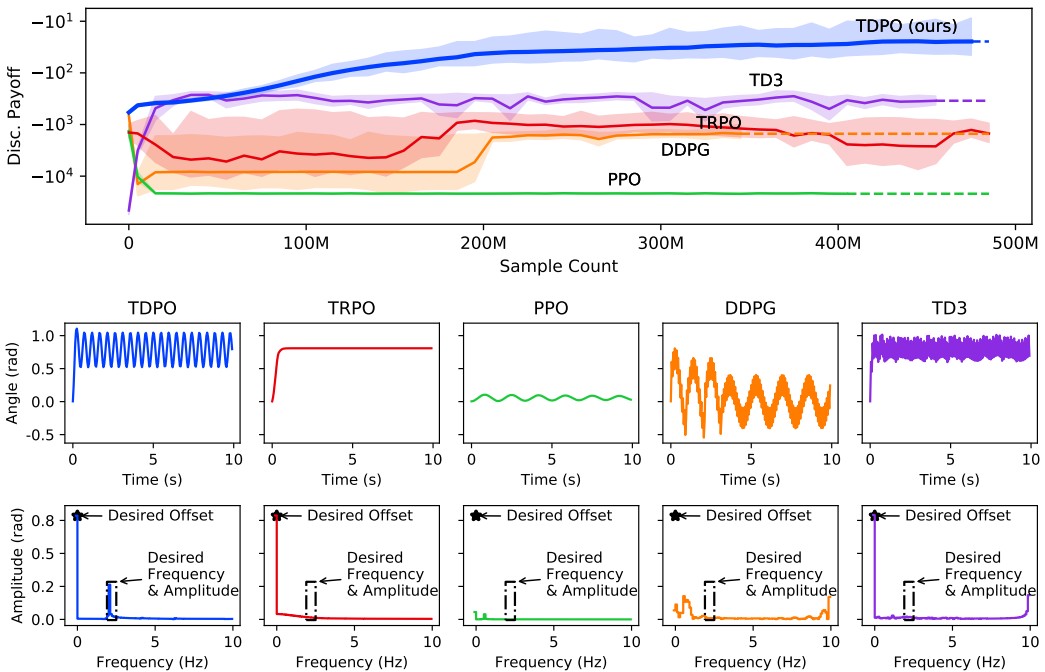

Figure 7: Results for the fourth variant of the simple pendulum with non-local rewards. Upper panel: training curves with empirical discounted payoffs. Lower panels: trajectories in both the time domain and frequency domain, showing target values of oscillation frequency, amplitude, and offset.

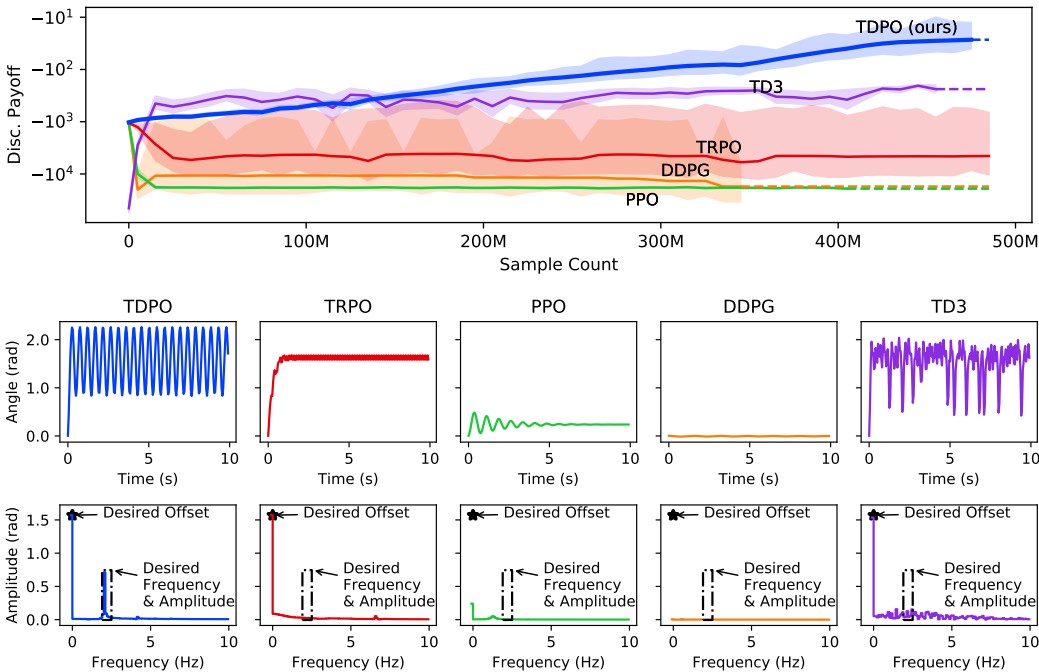

Figure 8: Results for the fifth variant of the simple pendulum with non-local rewards. Upper panel: training curves with empirical discounted payoffs. Lower panels: trajectories in both the time domain and frequency domain, showing target values of oscillation frequency, amplitude, and offset.

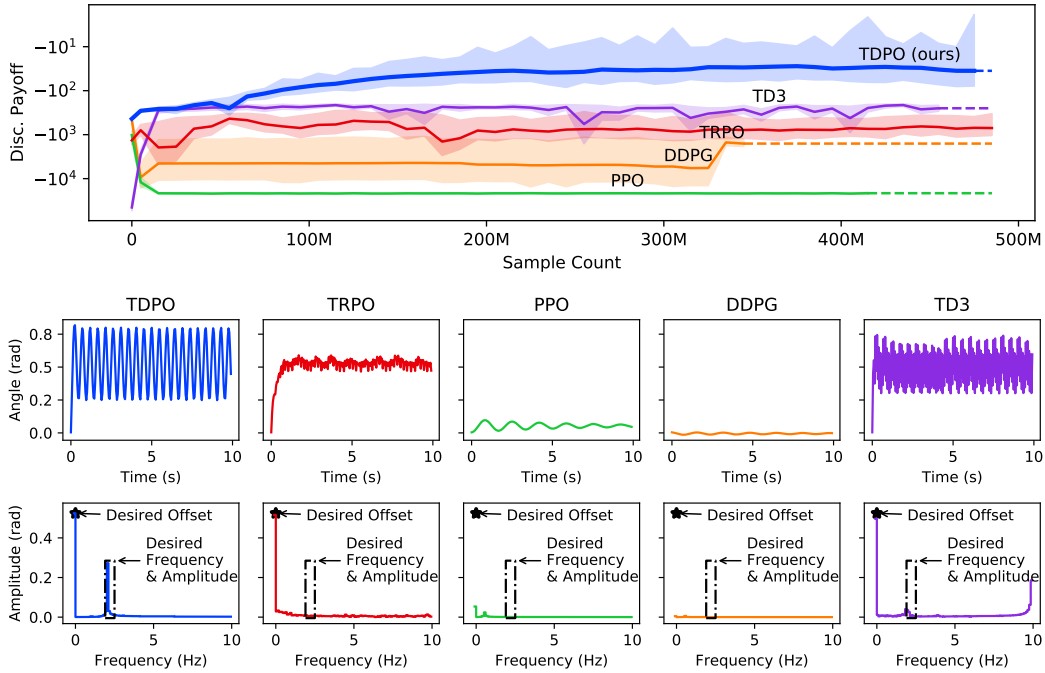

Figure 9: Results for the sixth variant of the simple pendulum with non-local rewards. Upper panel: training curves with empirical discounted payoffs. Lower panels: trajectories in both the time domain and frequency domain, showing target values of oscillation frequency, amplitude, and offset.

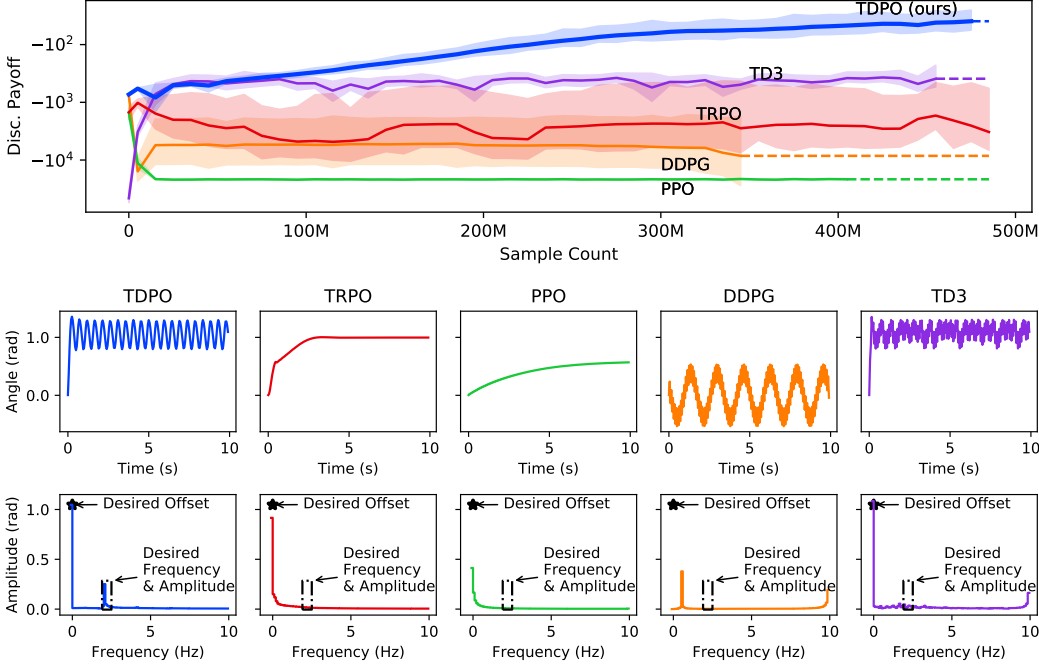

Figure 10: Results for the seventh variant of the simple pendulum with non-local rewards. Upper panel: training curves with empirical discounted payoffs. Lower panels: trajectories in both the time domain and frequency domain, showing target values of oscillation frequency, amplitude, and offset.

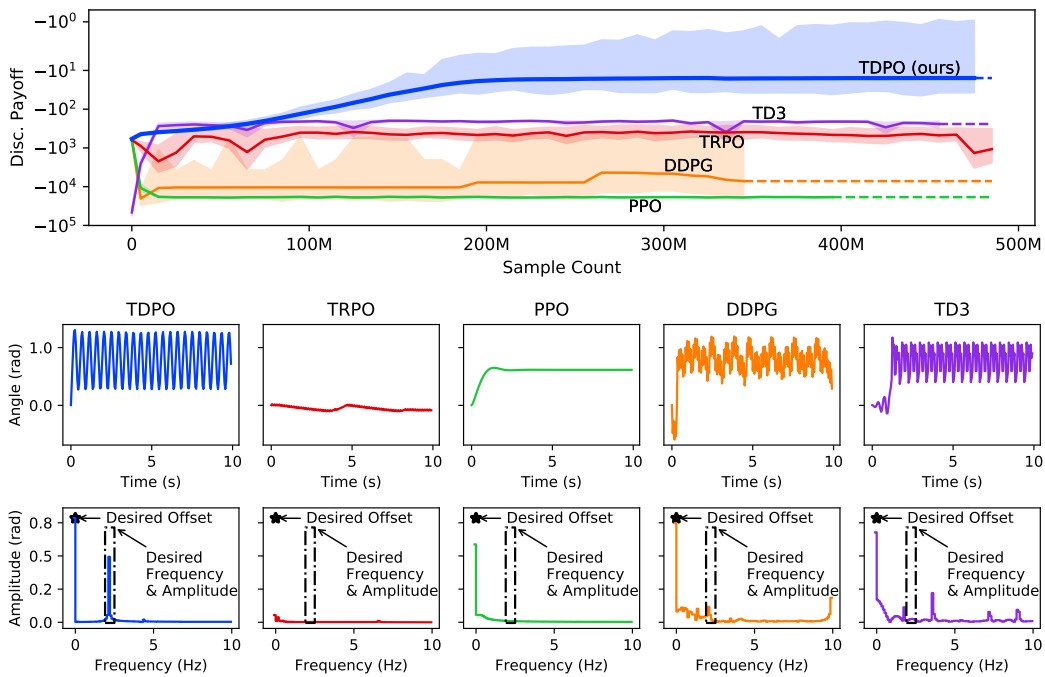

Figure 11: Results for the eighth variant of the simple pendulum with non-local rewards. Upper panel: training curves with empirical discounted payoffs. Lower panels: trajectories in both the time domain and frequency domain, showing target values of oscillation frequency, amplitude, and offset.

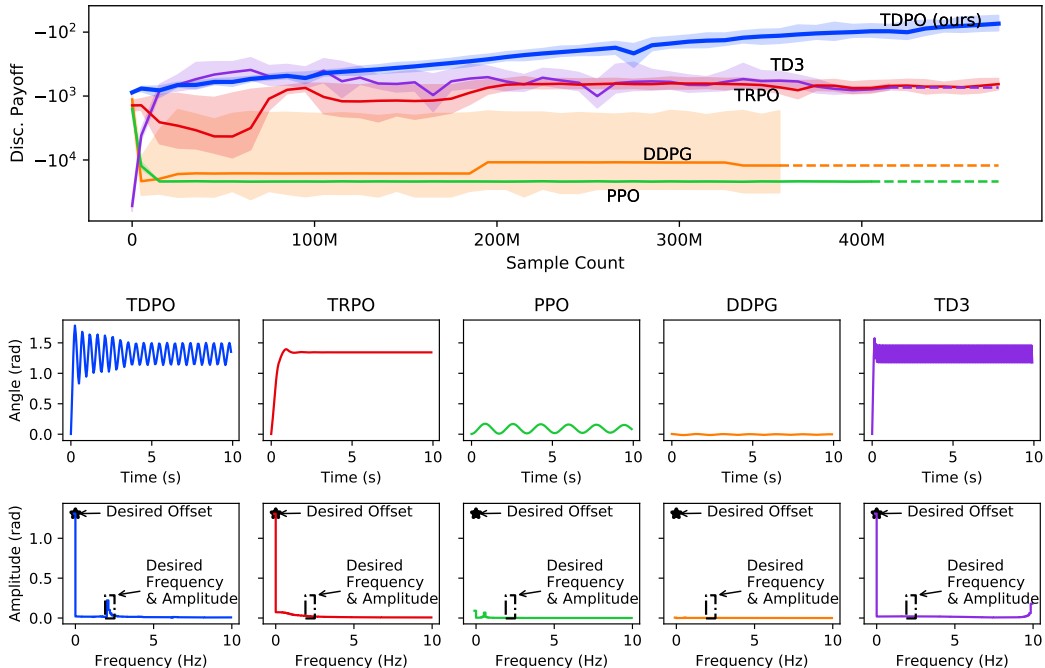

Figure 12: Results for the ninth variant of the simple pendulum with non-local rewards. Upper panel: training curves with empirical discounted payoffs. Lower panels: trajectories in both the time domain and frequency domain, showing target values of oscillation frequency, amplitude, and offset.

in Sections A.9, A.6, and A.10. We will describe how to implement TDPO, and focus on the subtle differences between TDPO and TRPO next.

As for the state-reset capability, our algorithm does not require access to a reset function to arbitrary states. Instead, we only require to be able to start from the prior trajectory's initial state. Many environments, including the Gym environments, instantiate their own pseudo-random generators and only utilize that pseudo-random generator for all randomized operations. This facilitates a straightforward implementation of the DeVine oracle; in such environments, implementing an arbitrary state-reset functionality is unnecessary, and only reloading the pseudo-random generator to its configuration prior to the trajectory would suffice. In other words, the DeVine oracle can store the initial configuration of the pseudo-random generator before asking for a trajectory reset, and then start sampling. Once the main trajectory is finished, the pseudo-random generator can be reloaded, thus producing the same initial state upon a reset request. Other time-step states can then be recovered by applying the same preceeding action sequence.

To optimize the quadratic surrogate, the conjugate gradient solver was used. Implementing the conjugate gradient algorithm is fairly straightforward, and is already included in many common automatic differentiation libraries. The conjugate gradient solver is perfect for situations where (1) the Hessian matrix is larger than can efficiently be stored in the memory, and (2) the Hessian matrix includes many nearly identical eigenvalues. Both of these conditions apply well for TDPO, as well as for TRPO. Instead of requiring the full Hessian matrix to be stored, the conjugate gradient solver only requires a Hessian-vector product machinery $v \to Hv$, which must be specifically implemented for TDPO. Our surrogate function can be viewed as

$$\mathcal{L}(\delta\theta) = g^T \delta\theta + \frac{C_2'}{2} \delta\theta^T H \delta\theta$$

where the Hessian matrix can be defined as

$$H = H_2 + \frac{C_1'}{C_2'} H_1,$$

$$H_1 := \nabla_{\theta'}^2 \mathbb{E}_{s \sim \rho_\mu^{\pi_k}} \left[ \mathcal{L}_{G^2}(\pi', \pi_k; s) \right],$$

$$H_2 := \nabla_{\theta'}^2 \mathbb{E}_{s \sim \rho_\mu^{\pi_k}} \left[ W(\pi'(a|s), \pi_k(a|s))^2 \right]. \tag{73}$$

In order to construct a Hessian-vector product machinery $v \to Hv$, one can design an automatic-differentiation procedure that returns the Hessian-vector product. Many automatic-differentiation packages already include functionalities that can provide a Hessian-vector product machinery of a given scalar loss function without computing the Hessian matrix. This can be used to implement the Hessian-vector product machinery in a straightforward manner; one only needs to provide the scalar quadratic terms of our surrogate, and would obtain the Hessian-vector product machinery in return. On the other hand, this may not be the most computationally efficient approach, as our problem exhibits a more specific structure. Alternatively, one can implement a more elaborate and specifically designed Hessian-vector product machinery by following these three steps:

- Compute the Wasserstein-vector product $v \to H_2 v$ according to Algorithm 3.
- Compute the Sensitivity-vector product $v \to H_1 v$ according to Algorithm 4.
- Return the weighted sum of $H_1 v$ and $H_2 v$ as the final Hessian-vector product $Hv$.

One may also need to add a conjugate gradient damping to the conjugate gradient solver (i.e., return $\beta v + Hv$ for some small $\beta$ as opposed to returning $Hv$ purely), which is also done in the TRPO method. This may be important when the number of policy parameters is much larger than the sample size. Setting $\beta = 0$ may yield poor numerical stability if $H$ had small eigenvalues, and setting large $\beta$ will cause the conjugate gradient optimizer to mimic the gradient descent optimizer by making updates in the same direction as the gradient. The optimal conjugate gradient damping may depend on the problem and other hyper-parameters such as the sample size. However, it can easily be picked to be a small value that ensures numerical stability.

Once the conjugate gradient solver returned the optimal update direction $H^{-1}g$, it must be scaled down by a factor of $C_2'$ (i.e., $\delta\theta^* = H^{-1}g/C_2'$). If $\delta\theta^*$ satisfied the trust region criterion (i.e.,

$\frac{1}{2}\delta\theta^{*T}H\delta\theta^* \leq \delta_{\max}^2$), then one can make the parameter update (i.e., $\theta_{\text{new}} = \theta_{\text{old}} + \delta\theta^*$) and proceed to the next iteration. Otherwise, the proposed update $\delta\theta^*$ must be scaled down further, namely by $\alpha$, such that the trust region condition would be satisfied (i.e., $\frac{1}{2}(\alpha\delta\theta^*)^T H(\alpha\delta\theta^*) = \delta_{\max}^2$) before making the update $\theta_{\text{new}} = \theta_{\text{old}} + \alpha\delta\theta^*$.

---

**Algorithm 3** Wasserstein-Vector-Product Machinery

---

**Require:** Current Policy $\pi_1$ with parameters $\theta_1$.
**Require:** The vector $v$ with the same dimensions as $\theta_1$.
**Require:** An observation $s$.
 1: Compute the action for the observation $s$ with $|A|$ elements.

$$a_{|A|\times 1} := \begin{bmatrix} \pi^{(1)}(s) \\ \vdots \\ \pi^{(|A|)}(s) \end{bmatrix}. \tag{74}$$

   This vector should be capable of propagating gradients back to the policy parameters when used in automatic differentiation software.
 2: Define $t$ to have be a constant vector with the same shape as $a$. It could be populated with any values such as all ones.
 3: Define the scalar $\tilde{a} := a^T t$.
 4: Using back-propagation, find the gradient

$$\nabla_\theta \tilde{a} = \sum_{i=1}^{|A|} t_i \nabla_\theta a_i = \sum_{i=1}^{|A|} t_i \begin{bmatrix} \frac{\partial a_i}{\partial \theta_1} & \cdots & \frac{\partial a_i}{\partial \theta_{|\Theta|}} \end{bmatrix}. \tag{75}$$

 5: Compute the following dot-product:

$$\langle \nabla_\theta \tilde{a}, v \rangle = \left( \sum_{i=1}^{|A|} t_i \cdot \frac{\partial a_i}{\partial \theta_1} \right) \cdot v_1 + \cdots + \left( \sum_{i=1}^{|A|} t_i \cdot \frac{\partial a_i}{\partial \theta_{|\Theta|}} \right) \cdot v_{|\Theta|}. \tag{76}$$

 6: Using automatic differentiation, take the gradient w.r.t. the $t$ vector.

$$\tilde{a}_{\theta,v} := \nabla_t \langle \nabla_\theta \tilde{a}, v \rangle = \begin{bmatrix} \frac{\partial a_1}{\partial \theta_1} \cdot v_1 + \cdots + \frac{\partial a_1}{\partial \theta_{|\Theta|}} \cdot v_{|\Theta|} \\ \vdots \\ \frac{\partial a_{|A|}}{\partial \theta_1} \cdot v_1 + \cdots + \frac{\partial a_{|A|}}{\partial \theta_{|\Theta|}} \cdot v_{|\Theta|} \end{bmatrix} = \begin{bmatrix} \frac{\partial a_1}{\partial \theta_1} & \cdots & \frac{\partial a_1}{\partial \theta_{|\Theta|}} \\ \vdots & & \vdots \\ \frac{\partial a_{|A|}}{\partial \theta_1} & \cdots & \frac{\partial a^{|A|}}{\partial \theta_{|\Theta|}} \end{bmatrix} v \tag{77}$$

 7: Compute the dot product $\langle \tilde{a}_{\theta,v}, \tilde{a} \rangle$.
 8: Using back-propagation, take the gradient w.r.t. $\theta$, and return it as the gain-vector-product.

$$\nabla_\theta \langle \tilde{a}_{\theta,v}, \tilde{a} \rangle = \begin{bmatrix} \frac{\partial a_1}{\partial \theta_1} & \cdots & \frac{\partial a_1}{\partial \theta_{|\Theta|}} \\ \vdots & & \vdots \\ \frac{\partial a_{|A|}}{\partial \theta_1} & \cdots & \frac{\partial a^{|A|}}{\partial \theta_{|\Theta|}} \end{bmatrix}^T \begin{bmatrix} \frac{\partial a_1}{\partial \theta_1} & \cdots & \frac{\partial a_1}{\partial \theta_{|\Theta|}} \\ \vdots & & \vdots \\ \frac{\partial a_{|A|}}{\partial \theta_1} & \cdots & \frac{\partial a^{|A|}}{\partial \theta_{|\Theta|}} \end{bmatrix} v \tag{78}$$

---

---

**Algorithm 4** Sensitivity-Vector-Product Machinery

---

**Require:** Current Policy $\pi_1$ with parameters $\theta_1$.
**Require:** The vector $v$ with the same dimensions as $\theta_1$.
**Require:** An observation $s$.
1: Compute the action to observation Jacobian matrix

$$J_{|A|\times|S|} := \begin{bmatrix} \frac{\partial \pi^{(1)}(s)}{\partial s^1} & \cdots & \frac{\partial \pi^{(1)}(s)}{\partial s^{(|S|)}} \\ \vdots & & \vdots \\ \frac{\partial \pi^{(|A|)}(s)}{\partial s^{(1)}} & \cdots & \frac{\partial \pi^{(|A|)}(s)}{\partial s^{|S|}} \end{bmatrix}. \tag{79}$$

This can either be done using finite-differences in the observation using

$$\frac{\partial \pi^{(i)}(s)}{\partial s^{(j)}} \simeq \frac{\pi^{(i)}(s + ds \cdot \mathbf{e_j}) - \pi^{(i)}(s)}{ds} \tag{80}$$

(which may be a bit numerically inaccurate), or using automatic differentiation. In any case, this matrix should be a parameter tensor capable of propagating gradients back to the parameters when used in automatic differentiation software.
2: Define $\tilde{J}$ to be the vectorized (i.e., reshaped into a column) $J$ matrix, with $|AS| = |A| \times |S|$ rows and one column.
3: Define $t$ to have be a constant vector with the same shape as $\tilde{J}$. It could be populated with any values such as all ones.
4: Define the scalar $J_t := \tilde{J}^T t$.
5: Using back-propagation, find the gradient

$$\nabla_\theta J_t = \sum_{i=1}^{|A|} \sum_{j=1}^{|S|} t_{i,j} \nabla_\theta J_{i,j} = \sum_{i=1}^{|A|} \sum_{j=1}^{|S|} t_{i,j} \begin{bmatrix} \frac{\partial J_{i,j}}{\partial \theta_1} & \cdots & \frac{\partial J_{i,j}}{\partial \theta_{|\Theta|}} \end{bmatrix}. \tag{81}$$

6: Compute the following dot-product.

$$\langle \nabla_\theta J_t, v \rangle = (\sum_{i=1}^{|A|} \sum_{j=1}^{|S|} t_{i,j} \cdot \frac{\partial J_{i,j}}{\partial \theta_1}) \times v_1 + \cdots + (\sum_{i=1}^{|A|} \sum_{j=1}^{|S|} t_{i,j} \cdot \frac{\partial J_{i,j}}{\partial \theta_{|\Theta|}}) \times v_{|\Theta|} \tag{82}$$

7: Using automatic differentiation, take the gradient w.r.t. the $t$ vector.

$$(\nabla_\theta J)v := \nabla_t \langle \nabla_\theta J_t, v \rangle = \begin{bmatrix} \frac{\partial J_{1,1}}{\partial \theta_1} \cdot v_1 + \cdots + \frac{\partial J_{1,1}}{\partial \theta_{|\Theta|}} \cdot v_{|\Theta|} \\ \vdots \\ \frac{\partial J_{|A|,|S|}}{\partial \theta_1} \cdot v_1 + \cdots + \frac{\partial J_{|A|,|S|}}{\partial \theta_{|\Theta|}} \cdot v_{|\Theta|} \end{bmatrix}$$

$$= \begin{bmatrix} \frac{\partial J^{(1,1)}}{\partial \theta_1} & \cdots & \frac{\partial J^{(1,1)}}{\partial \theta_{|\Theta|}} \\ \vdots & & \vdots \\ \frac{\partial J^{(|A|,|S|)}}{\partial \theta_1} & \cdots & \frac{\partial J^{(|A|,|S|)}}{\partial \theta_{|\Theta|}} \end{bmatrix} v \tag{83}$$

8: Reshape $(\nabla_\theta J)v$ into a column vector and name it $\tilde{J}_{\theta,v}$.
9: Compute the dot product $\langle \tilde{J}_{\theta,v}, \tilde{J} \rangle$.
10: Using back-propagation, take the gradient w.r.t. $\theta$, and return it as the gain-vector-product.

$$\nabla_\theta \langle \tilde{J}_{\theta,v}, \tilde{J} \rangle = \begin{bmatrix} \frac{\partial J^{(1,1)}}{\partial \theta_1} & \cdots & \frac{\partial J^{(1,1)}}{\partial \theta_{|\Theta|}} \\ \vdots & & \vdots \\ \frac{\partial J^{(|A|,|S|)}}{\partial \theta_1} & \cdots & \frac{\partial J^{(|A|,|S|)}}{\partial \theta_{|\Theta|}} \end{bmatrix}^T \begin{bmatrix} \frac{\partial J^{(1,1)}}{\partial \theta_1} & \cdots & \frac{\partial J^{(1,1)}}{\partial \theta_{|\Theta|}} \\ \vdots & & \vdots \\ \frac{\partial J^{(|A|,|S|)}}{\partial \theta_1} & \cdots & \frac{\partial J^{(|A|,|S|)}}{\partial \theta_{|\Theta|}} \end{bmatrix} v \tag{84}$$

---

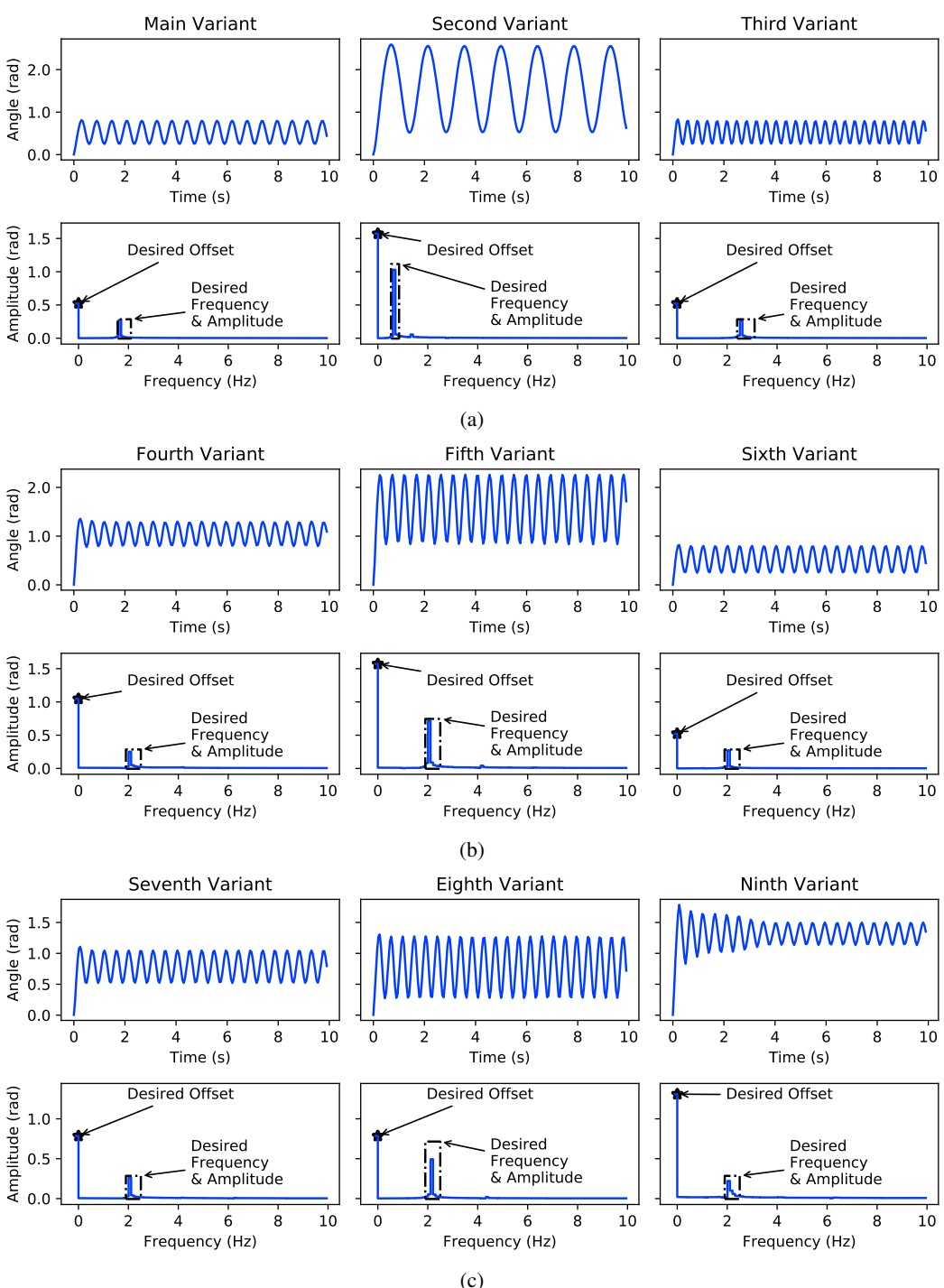

Figure 13: Time and frequency domain trajectories for our method (TDPO) on multiple variants of the simple pendulum with non-local rewards. (a) The high-reward trajectories for the first group of variants, (b) the high-reward trajectories for the second group of variants, (c) the high-reward trajectories for the third group of variants. Target values of oscillation frequency, amplitude, and offset were annotated in the frequency domain plots.

