# OpenReview forum: "Truly Deterministic Policy Optimization"
_ICLR.cc/2021/Conference — Reject_

### Official Review · AnonReviewer3 · 2020-10-27
**A variant of TRPO for deterministic policy**

**Rating:** 5
**Confidence:** 3

**Review:**

This paper builds on the template of TRPO and proposes an algorithm that searches for deterministic policy. A novel policy improvement lemma is derived for Wasserstein distance and thus a new algorithm is proposed based on iteratively optimizing policy performance lower bound.

I feel overall this paper may not be quite well motivated.  When describing the disadvantage of stochastic policies, it is better to make it clear. It is hard to get why stochastic policies are not able to handle non-local rewards, long time horizons, and naturally-resonant environments, in what aspects? What exactly means non-local rewards? Why TRPO can not handle long time horizons? There is no citation for "exploratory noise injection"? Where does the noise be injected? Many terms used here may lack of precise definitions. Overall I suggest the authors to make a clear comparison with TRPO that in what aspect you can improve TRPO. This is better to present the novelty of current works. Some improvements on MuJoCo over TRPO might not be sufficiently enough since RL algorithms involve too much tricks to make the comparison completely fair.

In (9), the theoretical lower bound to be optimized involves C_1 and C_2 that has a very complicated form. For the practical algorithm, I feel you just treat them as a tuning parameter when translating the regularized form to constrained form. Will it be problematic? In Section 5.1, the modification of reward function for the pendulum is a bit artificial. Anyone did this before or what's the reason to do that? Again, this may correspond the motivation of the proposed algorithm.

The presentation of Section 2.1 may not be in an optimal way. This should be a main result but several key derivations (Inequality (31) and Theorems A.5 and A.4) are deferred to the appendix.

---

> ### Author Response · Authors · 2020-11-21
> **Third Reviewer's Response (Part 2)**
>
> * *Improving the Background Section*: We acknowledge the density of the background section, which makes it difficult to follow. To alleviate this, we added two tables of notation to the appendix summarizing the same mathematical definitions and notations. We also have referenced these two tables in Section 1 of the paper.
>
> **Refrences**:
>
> [1] Liu, Qiang, et al. "Breaking the curse of horizon: Infinite-horizon off-policy estimation." *Advances in Neural Information Processing Systems*. 2018.
>
> [2] Kakade, Sham Machandranath. *On the sample complexity of reinforcement learning*. Diss. University of London, 2003.
>
> [3] Jiang, Nan, and Alekh Agarwal. "Open problem: The dependence of sample complexity lower bounds on planning horizon." *Conference On Learning Theory*. 2018.
>
> [4] Dann, Christoph, and Emma Brunskill. "Sample complexity of episodic fixed-horizon reinforcement learning." *Advances in Neural Information Processing Systems*. 2015.
>
> [5] Kearns, Michael J., Yishay Mansour, and Andrew Y. Ng. "Approximate planning in large POMDPs via reusable trajectories." *Advances in Neural Information Processing Systems*. 2000.
>
> [6] Kearns, Michael, Yishay Mansour, and Andrew Y. Ng. "A sparse sampling algorithm for near-optimal planning in large Markov decision processes." *Machine learning* 49.2-3 (2002): 193-208.
>
> [7] Wang, Ruosong, et al. "Is Long Horizon Reinforcement Learning More Difficult Than Short Horizon Reinforcement Learning?." *arXiv preprint arXiv:2005.00527* (2020).
>
> [8] Kallus, Nathan, and Masatoshi Uehara. "Efficiently Breaking the Curse of Horizon in Off-Policy Evaluation with Double Reinforcement Learning." *arXiv* (2019): arXiv-1909.
>
> [9] Kallus, Nathan, and Masatoshi Uehara. "Statistically efficient off-policy policy gradients." *arXiv preprint arXiv:2002.04014* (2020).

---

> ### Author Response · Authors · 2020-11-21
> **Third Reviewer's Response (Part 1)**
>
> We thank the reviewer for the valuable comments, and will try to respond to the questions one by one.
>
> * *Formal Definition of Non-local Rewards*: We acknowledge the fact that a formal definition of the non-local rewards was missing. We have added a footnote definition to Section 5.1 which provides a mathematical description of non-local rewards. Non-local rewards are reward functions of the entire trajectory whose payoffs cannot be decomposed into the sum of terms such as $\eta=\sum_{t} f_t(s_t,a_t)$ where functions $f_t$ only depend on nearby states and actions. Examples of such rewards include $R(s_1,a_1,\cdots,s_T,a_T)=\max_t s_t$ and the reward that depends on the Fourier transform of the complete trajectory signal.
>
> * *What Noise Injection Means*: We acknowledge that the term "exploratory noise injection" was not clearly defined in the main text, and was left to be inferred from the context. We have added a paragraph in the introduction explaining what constant exploratory noise injection means. Stochastic policy search methods such as TRPO, PPO, DDPG, and TD3 sample their actions stochastically during training by taking the deterministic output of the policy network and adding stochastic noise samples to them for the sake of exploration. Since this happens during all time-steps of a trajectory, we call this process "constant exploratory noise injection".
>
> * *Why TRPO Cannot Scale to Longer Horizons*: We acknowledge that a proper discussion regarding scalability to longer horizons is missing from the paper. We have now added a paragraph to our introduction section discussing this point and the poor scalability of existing methods to longer horizons.
>
>   The poor performance of existing methods under long horizons is well-known in the community and is commonly referred to as the the "curse of horizon" in reinforcement learning ([1]). General worst-case analyses suggests that the sample complexity of reinforcement learning is exponential with respect to the horizon length ([2], [5], and [6]). Deriving polynomial lower-bounds for the sample complexity of reinforcement learning methods is still an open problem ([3]). Lower-bounding the sample complexity of reinforcement learning for long horizons under different settings and simplifying assumptions has been a topic of theoretical research ([4] and [7]). Some recent work has closely examined the importance sampling gradient estimators scaling to horizon in terms of both theoretical and practical estimation variances ([1], [8], and [9]). All in all, long horizons certainly impose scalability challenges for all reinforcement learning methods, especially the ones suffering from excessive estimation variance due to the stochastic policy search procedures employed within them. Our method does not suffer from this curse in the same way because it does not induce exploding variances as the horizon increases.
>
> * *Why Deterministic Policies Are Preferable on Non-local Rewards*: Stochastic policy optimization methods (i.e., TRPO, PPO, DDPG, and TD3) may not perform as well as our deterministic policy search method. Performing _deterministic search_ using the DeVine oracle could be advantageous in the presence of non-local rewards; the DeVine oracle is insensitive to whether the rewards were applied immediately or at the final time-step of the trajectory, since it only needs the difference of the two trajectory payoffs. On the other hand, all stochastic policy search methods (including DDPG and TD3) assume the environment provides an immediate reward at each time-step. The fact that (1) the intermediate time-step rewards are fixed at zero regardless of the actions taken by the agent, and (2) all the actions determine a single reward applied at the final time-step, may impact stochastic methods (i.e., TRPO, PPO, DDPG, and TD3) advantage estimation oracles negatively since their algorithmic design relies on immediate rewards being applied at each time step.
>
> * *Choice of $C_1$ and $C_2$*: In this initial version, we treated the regularization coefficients $C_1$ and $C_2$ as constant hyper-parameters, and fixed them throughout training. We have described the specific choices of these coefficients for each environment in Section A.6 of the appendix, where we have provided intuition and some rules of thumb for how to pick these coefficients. The trust region machinery helps mitigate any over-sensitivity to these coefficients. Adaptively picking the regularization coefficients $C_1$ and $C_2$ is an excellent topic for future research, which could improve the performance of our method significantly.

---

### Official Review · AnonReviewer4 · 2020-10-28
**Blind Review #4**

**Rating:** 6
**Confidence:** 4

**Review:**

The paper proposes a deterministic policy gradient method using the Wasserstein distance to quantify the difference of deterministic policies. It also introduces a new deterministic estimator for the policy gradient with theoretical justification. The paper uses a similar path as TRPO to first introduce a surrogate objective then replace the complex term by a tractable term for practical performance. The method has been shown to work well under long-horizon and non-local reward environments whereas existing methods struggle to solve.

Strength:
- Being able to deal with non-traditional RL settings can be beneficial to the RL community.
- The adaptation of Wasserstein distance into the surrogate objective of TRPO is new.
- Theoretical justification for the main results.
- Numerical experiments show advantage of the proposed method in non-local or long-horizon settings.

Weakness:
- The presentation of background section can be improved as it is too dense with different notations. It may be a good idea to summarize these notations into a table.
- Only consider 2 environments: 1 with non-local reward and 1 with long horizon and resonant frequencies.

I have the following questions and comments:
Q1. What is the motivation of changing frequency of pendulum to 1.7? instead of 0.5Hz? Have you tried different settings of the pendulum environment (use the original frequency of 0.5Hz)? If yes I am curious how other methods perform compare to TDPO.

C1. It would be nice to have a discussion to compare the computation cost of each iteration of TDPO and other methods like TRPO and TD3.

C2. As shown in the experiments, TDPO does not work well in the common gym environments, is there any changes to algorithm design possible to improve the current algorithm for these setting?

C3. From the way the method is presented, it seems that it is not simple to implement the method? It is better to have a discussion on the implementation aspect of the method.

Small comments:
- Secion 1, first paragraph, line 6, there are two \delta_s
- The definition of KL divergence in terms of Hilbert space inner product missing a subscript?
- Introduce the definition of the term h.o.t. before use although it can be inferred from the context

Overall, I suggest a weak-accept decision based on the following reason:
- Novelty in algorithm development with theoretical justification for the monotonic improvement.
- Although experiments indeed show the clear advantage of the proposed method over existing ones, more environments or more setting in the same environment should be presented to better evaluate its performance.

---

> ### Author Response · Authors · 2020-11-21
> **Fourth Reviewer's Response**
>
> We thank the reviewer for the thoughtful and well-structured comments and suggestions which definitely improved our paper and made it stronger.
>
> We acknowledge the density of the background section, which makes it difficult to follow. To alleviate this, we added two tables of notation to the appendix summarizing the same mathematical definitions and notations. We also have referenced these two tables in Section 1 of the paper.
>
> **Question 1**. The reviewer is right to point out the missing study in our paper. We have now added 8 new variants of the same pendulum to cover many target signal characteristics. The results are shown in the new appendix Section A.13. The new environments include the second variant, which targets the 0.5 Hz frequency. In short, we find our method to be able to consistently produce the desired signal characteristics, while no other method was able to solve such environments effectively.
>
> **Comment 1**: We have now added a comparison of the different methods' running times to the paper's appendix (see Section A.12). In short, our method (TDPO) is the fastest among all (i.e., TRPO, PPO, DDPG, and TD3). This is mainly due to the use of the DeVine gradient estimator which summarizes each two trajectories into a single state-action-advantage triple. This significantly cuts down on the optimization cost of our algorithm.
>
> **Comment 2**: There are multiple strategies designed within methods such as TRPO and PPO that could presumably benefit our method. Unfortunately, we have not yet had time to investigate all of these strategies. In particular, the regularization coefficients $C_1, C_2$ are set to be pre-determined constants throughout training which is likely to be inefficient. Also, we have not explored different kinds of network initializations, layer scalings, or architectural changes. We did not perform any kind of observation, action, or reward normalizations or clippings, as well. Furthermore, using different exploration scales or regularization coefficients for each action dimension separately could substantially improve our performance on multi-dimensional action spaces of the gym suite. We could also pause sampling within trajectories to make a parameter update and then continue the trajectory to boost the sample efficiency, as other methods do. Annealing hyper-parameters and adding intuitive terms to the optimization surrogate may also be helpful (similar to the entropy term in PPO's objective). Such strategies are used in other methods and could improve our sample efficiency on these environments.
>
> **Comment 3**: The reviewer is correct that implementing our method requires some effort. We have now added Section A.14 to the appendix, which discusses the implementation aspect of our method in detail.  Broadly speaking, implementing our method (TDPO) is very similar in complexity to TRPO; both employ the conjugate gradient optimization method using vector-product oracles. However, TRPO performs line-search heuristics, which is absent in our method.
>
> We thank the reviewer again for the constructive comments, which definitely raised the quality of our empirical studies and made the proper stronger and more informative.

---

### Official Review · AnonReviewer2 · 2020-10-29
**This paper introduces a policy gradient method based on deterministic policies and deterministic gradient estimates. The authors show that the proposed technique can estimate gradients on long-horizon tasks without the need to inject noise into the system for exploration. Even though the experiments may suggest that the method is not widely applicable to general MDPs, I believe that the underlying theory is, on its own, an interesting contribution.**

**Rating:** 6
**Confidence:** 3

**Review:**

This paper introduces a policy gradient method based on deterministic policies and deterministic gradient estimates. By assuming such a deterministic setting, the authors show that the proposed technique can estimate gradients on long-horizon tasks without the need to inject noise into the system for exploration. The authors propose using a Wasserstein-based model to perform regularization in this particular setting involving deterministic policies. They also establish the conditions under which it is possible to guarantee monotonic policy improvement.

Overall, this is a well-written paper with sound mathematical arguments. The authors present a convincing review of related work and argue that the proposed method is better-suited than existing techniques when dealing with problems where stochastic exploration is not efficient or feasible. The authors formally show that it is possible to lower bound the payoff improvement, which is a non-trivial result. All update equations are carefully derived and discussed in the appendix.

The experiments seem to show that the method works well in a possibly restricted set of settings, involving either control tasks with resonant frequencies or problems with rewards associated with achieving oscillatory behaviors. The experiments on these particular handcrafted problems are well-designed and demonstrate that the proposed method (TDPO) achieves better performance than its competitors. However, the proposed method generally performs worse (on average) than others when evaluated on standard control benchmarks. Even though the experiments may suggest that the method is not widely applicable, I believe that the underlying theory is, on its own, an interesting contribution.

I believe that this paper introduces a relevant contribution to the RL community that is concerned with scaling up learning to long-horizon problems. It presents a principled method and introduces non-trivial bounds. In my opinion, this conference's community would benefit from having this paper accepted to its proceedings.


I have a few questions for the authors:

1) in the introduction, the authors discuss rewards defined in the frequency domain. Could you please clarify what those are, in terms of the corresponding structure that they may imply in the reward function?

2) the authors also argue that the proposed method is robust in control environments with resonant frequencies. Could you formally define what this means, and what are the implications of this assumption when deploying this method in more general MDPs which may not be modeling robotics problems?

3) in the Background section, P is defined as DeltaS x DeltaA -> DeltaS. Shouldn't it be SxA -> DeltaS, instead?

4) in the Background section, there is an issue with the sentence that begins with "We may abuse notation and replace (...)". I believe that the terms mentioned here should have been delta_s, delta_a, s, a.

5) notation: P is sometimes used to refer to the transition dynamics, but sometimes it refers to the probability of an event. This isn't very clear. See, e.g., the reuse of this symbol when defining P(mu_s, pi).

6) what is P(mu_s, pi(. | s~mu_s))? Is this the joint distribution over mu_s and pi? If "P" denotes a probability, I'm confused: the terms used at this point do not refer to random variables or events, but to distributions. Could you please improve the notation used when defining such a generalized dynamics?

7) you assume the existence of Lip(Q^pi; a). Could you please discuss a few practical problems (with continuous actions) where the Q-function does not vary smoothly with actions? A discussion on the practical implications of this assumption would be interesting.

8) what is the intuition behind Eqs. 5 and 6? What do they mean, intuitively, and how do they relate to the goal of regularizing deterministic policies, and to the goal of ensuring monotonic policy improvement? These seem to be key assumptions to the method, but (in my opinion) they were not well motivated or discussed.

9) in this paper, your focus is on robotic environments with state-reset capabilities. This might be an unrealistic assumption, particularly in robotics problems where the goal is precisely to autonomously learn how to reach particular states or configuration. This assumption might also be reflecting deeper requirements; e.g., that the tasks being tackled are not goal-achieving tasks. Could you please discuss this assumption and its implications in more detail?

10) in practice, how do you pick the coefficients C_1 and C_2 (Eq.13)? How sensitive is your method to this choice?

---

> ### Author Response · Authors · 2020-11-21
> **Second Reviewer's Response (Part 3)**
>
> **Question 10**: In this initial version, we treated the regularization coefficients $C_1$ and $C_2$ as constant hyper-parameters, and fixed them throughout training. We have described the specific choices of these coefficients for each environment in A.6 of the appendix, where we have provided intuition and some rules of thumb for how to pick these coefficients. The trust region machinery helps mitigate any over-sensitivity to these coefficients. Adaptively picking the regularization coefficients $C_1$ and $C_2$ is an excellent topic for future research, which could improve the performance of our method significantly.
>
> We thank the reviewer again for all the valuable comments and questions, and sincerely appreciate the reviewer's careful examination of derivations, proofs, and assumptions which are the backbone of our algorithm.
>
> **References**:
>
> [1] Rachelson, Emmanuel and Lagoudakis, Michail G.. On the Locality of Action Domination in Sequential Decision Making. (2010) In: 11th International Symposium on Artificial Intelligence and Mathematics (ISIAM 2010).
>
> [2] Asadi, Kavosh, Dipendra Misra, and Michael L. Littman. "Lipschitz continuity in model-based reinforcement learning." *arXiv preprint arXiv:1804.07193* (2018).
>
> [3] Ha, Sehoon, Joohyung Kim, and Katsu Yamane. "Automated deep reinforcement learning environment for hardware of a modular legged robot." *2018 15th International Conference on Ubiquitous Robots (UR)*. IEEE, 2018.
>
> [4] Kuo, Benjamin C., and Farid Golnaraghi. "Automatic control systems." *Englewood cliffs* (2003).
>
> [5] Pirotta, Matteo, Marcello Restelli, and Luca Bascetta. "Policy gradient in lipschitz markov decision processes." *Machine Learning* 100.2-3 (2015): 255-283.

---

> ### Author Response · Authors · 2020-11-21
> **Second Reviewer's Response (Part 2)**
>
> **Question 7**: The reviewer brings up a natural concern where a proper discussion is missing from the paper. We have now added Section A.5.1 to the appendix, which is referenced in Section 2 of the main text and discusses the existence of the  $\text{Lip}(Q^\pi,a)$ constant and the mitigation factors.
>
> The $\text{Lip}(Q^\pi,a)$ constant may be undefined when either the reward function or the transition dynamics are discontinuous. Examples of known environments with undefined $\text{Lip}(Q^\pi,a)$ constants include those with grazing contacts which define a discontinuous transition dynamics. In practice, even for environments that do not satisfy Lipschitz continuity assumptions, there are mitigating factors; $Q^\pi$ functions are reasonably narrow-bounded in a small trust region neighborhood, and since we use non-vanishing exploration scales, a bounded interpolation slope can still model the $Q$-function variation effectively. We should also note that a slightly stronger version of this assumption is frequently used in the context of Lipschitz MDPs ([5], [1], and [2]). In practice, we have not found this to be a substantial limitation.
>
> **Question 8**: The reviewer brings up another natural concern over Assumptions 5 and 6, which were not properly discussed in the main paper. We have now added Section A.5.2 to the appendix to discuss these assumptions, and referenced it in the main paper as well.
>
> Assumptions 5 and 6 resemble the Lipschitz continuity assumptions of the transition dynamics with respect to actions and states, respectively. If the transition dynamics and the policy are deterministic, then these assumptions are exactly equivalent to the Lipschitz continuity assumptions. Assumptions 5 and 6 only generalize the Lipschitz continuity assumptions in a distributional sense.
>
> The necessity of these assumptions is a consequence of using metric measures for bounding errors. Traditional non-metric bounds force the use of full-support stochastic policies where all actions have non-zero probabilities (e.g., for the KL-divergence of two policies to be defined, TRPO needs to operate on full-support policies such as Gaussian policies). In those analyses, since all policies share the same support, the next state distribution automatically became smooth and Lipschitz continuous w.r.t. the policy measure even if the transition dynamics was not originally smooth w.r.t. its input actions. However, metric measures are also defined for policies of non-overlapping support. To be able to provide closeness bounds for future state visitations of two similar policies with non-overlapping support, it becomes necessary to assume that close-enough actions or states must be yielding close-enough next states. In fact, this is a very common assumption in the framework of Lipschitz MDPs (See Section 2.2 of [1], Section 3 of [2], and Assumption 1 of [5]).
>
> **Question 9**: The reviewer raises an important point. We have added information to the second paragraph of the new appendix section A.14. We also have replaced the "state-reset capability" requirement with a more precise "reset capability to previously visited states" in the introduction section. In fact, we do not require state-reset capability to arbitrary states. Instead, we particularly require the environment to be able to visit one of the states in the previous trajectory. (This could further be relaxed to only requiring to reproduce the initial state of the previous trajectory for deterministic dynamics.) Considering this clarification, such a requirement does not prohibit our method from being applied to goal-achieving tasks, and only prohibits it from being directly trainable on hardware. We acknowledge that there are efforts to train robotic agents using stochastic policy optimization methods directly on hardware (e.g., [3]). However, the vast majority of reinforcement learning robotics use simulated environments where such state-resets are quite possible.

---

> ### Author Response · Authors · 2020-11-21
> **Second Reviewer's Response (Part 1)**
>
> We thank the reviewer for the all the valuable and well-structured comments and questions, which certainly improved our paper and made it stronger. We will respond to the questions one by one.
>
> **Question 1**: The reviewer is right to point out the missing formal definition of non-local rewards. We have now added a footnote definition to Section 5.1's title, where we describe non-local rewards. Non-local rewards are reward functions of the entire trajectory whose payoffs cannot be decomposed into the sum of terms such as $\eta=\sum_{t} f_t(s_t,a_t)$ where functions $f_t$ only depend on nearby states and actions. Examples of such rewards include $\eta=R(s_1,a_1,\cdots,s_T,a_T)=\max_t s_t$ and the reward that depends on the Fourier transform of the complete trajectory signal.
>
> **Question 2**: The reviewer raises the legitimate concern that resonant frequencies are not formally defined in the paper. We have now added a footnote definition paragraph to the title of Section 5.2, where we describe what we mean by resonant frequencies. Resonant frequencies are concepts from control theory. In the frequency domain, signals of certain frequencies are excited more than others when applied to a system. This is captured by the frequency-domain transfer function of the system, which may have a peak of magnitude greater than one. The resonant frequency is the frequency at which the frequency-domain transfer function has the highest amplitude. Common examples of systems with a resonant frequency include the undamped pendulum, which oscillates at its natural frequency, and RLC circuits with characteristic frequencies at which they are most excitable. See chapter 8 of [4] for more information.
>
> Both discrete- and continuous-time systems can be analyzed in the frequency domain either using Fourier transformations in the continuous case or using Z-transformations in the discrete case. While frequency analysis is mostly common for robotics control environments in reinforcement learning, its application and insights may not be limited to robotic problems. The reward function of an MDP can be viewed as a system for instance; some rewards may heavily penalize oscillations of certain frequencies. Take the "change of direction" cost in a random walk for instance; a flat action signal would not incur any costs while oscillating behavior may be heavily penalized.
>
> **Question 3**: We thank the reviewer for the suggestion and have applied the correction. The standard definition of transition dynamics has the form $P: \mathcal{S}\times\mathcal{A}\to\Delta(\mathcal{S})$ as the reviewer suggested. We have now replaced the non-standard definition with the standard one, and described the other generalizations in the same paragraph. We have also added two tables of notation to the appendix, where we focused specifically on the multiple kinds of notational abuse involving the transition dynamics. We have also improved the notation of the former generalized dynamics $P(\mu_s, \pi)$ to be $\mathbb{P}(\mu_s, \pi)$.
>
> **Question 4**: We thank the reviewer for spotting this typo, and corrected the new document to reflect this change.
>
> **Question 5**: We thank the reviewer for noting this, and acknowledge the confusion. We have improved the generalized transition dynamics notation to be $\mathbb{P}(\mu_s, \pi)$ which accepts pairs of state distribution and policy as its input, as opposed to the standard transition dynamics $P(s,a)$ which takes state and action pairs as input. We added a sentence to the first paragraph of Section 1 of the paper explaining that the transition dynamics $P$ is defined as an operator which produces a distribution over the state space for the next state $s' \sim P(s,a)$. We also have removed the invalid definition $P^t(\mu, \pi) := \Pr [s_{t}|s_0\sim \mu(\cdot), a_k\sim\pi(\cdot|s_k)]$ from Lemma A.2 of the appendix, which was both incorrect and confusing. By removing this equation, our derivation became entirely free of event and random variable probability expressions (i.e., we do not use $\Pr[\cdot]$ anywhere in the derivation anymore and we did not intend to). We have also improved the notation of $P(\mu_s, \pi)$ to be $\mathbb{P}(\mu_s, \pi) := \mathbb{E}_{s\sim \mu_s}[\mathbb{E}_{a\sim \pi(s)}[P(s,a)]]$ (In other words, we are defining one distribution as an average over other distributions).
>
> **Question 6**: We thank the reviewer for making the notation improvement suggestion. We have improved the notation of the generalized transition dynamics and made it different from the standard transition dynamics. Now, $\mathbb{P}(\mu_s,\pi)$ defines the generalized transition dynamics and is never referred to as a probability of a random variable in the new revision. Please see the response to the previous question, which relates to the same issue.

---

### Official Review · AnonReviewer1 · 2020-10-30
**Interesting theoretical results but not convincing experiments.**

**Rating:** 5
**Confidence:** 3

**Review:**

This paper proposes a new deterministic policy gradient method (TDPO). The main idea and its derivations are based on the use of a deterministic Vine (DeVine) gradient estimator and the Wasserstein metric. The paper shows that a closed-form computation of Wasserstein distance can be derived without any approximation on the deterministic policies. A monotonic policy improvement guarantee is provided. Deterministic Vine (DeVine) is proposed as an optimization method for the proposed surrogate loss function, which is also inspired by the TRPO's surrogate loss. The proposed method is evaluated on two new robotic control domains that are proposed by the authors. The comparisons are against PPO, TRPO, DDPG, and TD3. Additional experiment results on Open AI Gym environments are provided in Appendix.

Overall, the studied idea is interesting. The finding of a closed-form Wasserstein distance on deterministic policies is useful. The new algorithm might be of interest to the RL community. Here are some of my main concerns regarding to this idea and the quality of the paper.

- The motivation of using the first experiment domain with non-local rewards is not clear on why this can other approaches have challenges? Why it could be a showcase of the use of a deterministic policy gradient method. In addition, as DDPG is an off-policy and deterministic policy gradient method, I was wondering why it does not perform well. More detailed experiment settings will be helpful.

- The second domain is basically a Mujoco task with a long-horizon. The results show it favors a deterministic method like TDPO. The same question on why DDPG and TD3 do performs well. They are off-policy methods that can be considered more sample-efficient than on-policy counter-parts like PPO/TRPO. On the other hand, much recent work show TRPO and PPO would perform comparably [1]. Therefore, a detailed experiment setting and a discussion on how each method's hypeparameters get tuned would be important.

- It is also questionable on why the TDPO does not perform well in comparisons to the baseline on Gym suite environments. Because those Gym domains are also based on Mujoco and programmed with deterministic dynamics.


[1] Logan Engstrom, Andrew Ilyas, Shibani Santurkar, Dimitris Tsipras, Firdaus Janoos, Larry Rudolph, Aleksander Madry:
Implementation Matters in Deep RL: A Case Study on PPO and TRPO. ICLR 2020

---

> ### Author Response · Authors · 2020-11-21
> **First Reviewer's Response (Part 2)**
>
> **Question 3**: We should note that the key difference between our environments and the gym suite is not the deterministic transition dynamics. We did not attempt to heavily optimize our own method, and avoided design decisions that could complicate the analysis. Instead, we have focused our attention on environments where the existing methods are performing poorly, and tried to address such fundamental shortcomings in a scalable algorithm. In fact, our method works so well on the non-locally rewarded and the long horizon environments that it is better anyways. To be competitive in other environments we still need to perform more optimization on our algorithm including (1) setting the regularization coefficients $C_1, C_2$ adaptively as opposed to being pre-determined constants throughout training, (2) exploring different kinds of network initializations, layer scalings, architectural modifications, etc., (3) performing observation, action, or reward normalizations or clippings, and (4) using different exploration scales for action dimension separately.
>
> **References**:
>
> [1] Liu, Qiang, Lihong Li, Ziyang Tang, and Dengyong Zhou. "Breaking the curse of horizon: Infinite-horizon off-policy estimation." *Advances in Neural Information Processing Systems*, 2018.
>
> [2] Kakade, Sham Machandranath. *On the sample complexity of reinforcement learning*. Diss. University of London, 2003.
>
> [3] Jiang, Nan, and Alekh Agarwal. "Open problem: The dependence of sample complexity lower bounds on planning horizon." *Conference On Learning Theory*. 2018.
>
> [4] Dann, Christoph, and Emma Brunskill. "Sample complexity of episodic fixed-horizon reinforcement learning." *Advances in Neural Information Processing Systems*. 2015.
>
> [5] Kearns, Michael J., Yishay Mansour, and Andrew Y. Ng. "Approximate planning in large POMDPs via reusable trajectories." *Advances in Neural Information Processing Systems*. 2000.
>
> [6] Kearns, Michael, Yishay Mansour, and Andrew Y. Ng. "A sparse sampling algorithm for near-optimal planning in large Markov decision processes." *Machine learning* 49.2-3 (2002): 193-208.
>
> [7] Wang, Ruosong, et al. "Is Long Horizon Reinforcement Learning More Difficult Than Short Horizon Reinforcement Learning?." *arXiv preprint arXiv:2005.00527* (2020).
>
> [8] Kallus, Nathan, and Masatoshi Uehara. "Efficiently Breaking the Curse of Horizon in Off-Policy Evaluation with Double Reinforcement Learning." *arXiv* (2019): arXiv-1909.
>
> [9] Kallus, Nathan, and Masatoshi Uehara. "Statistically efficient off-policy policy gradients." *arXiv preprint arXiv:2002.04014* (2020).

---

> ### Author Response · Authors · 2020-11-21
> **First Reviewer's Response (Part 1)**
>
> We thank the reviewer for the comments, and will respond to the raised concerns one by one.
>
> **Question 1**: We acknowledge the confusion when using similar terms such as "*Deterministic Search*" and "*Deterministic Policy Gradient Estimation*". We have added a paragraph at the introduction explaining that although DDPG and TD3 use deterministic gradient estimators, they are not *deterministic search* methods since they run stochastic policies to force exploration. We have also added another paragraph before Section 2 reinforcing the same subtle difference. Furthermore, we have included the formal definition of non-local rewards as a footnote at Section 5.1. In short, a non-local reward can only be defined after a full or partial trajectory was rolled out. In other words, in environments with non-local rewards, the intermediate time steps of the trajectory receive zero instant rewards, and a single non-zero reward is applied at the final time-step of the trajectory.
>
> As the reviewer suggested, there is no reason to believe that deterministic _gradient estimators_ are advantageous in the presence of non-local rewards. However, performing _deterministic search_ using the DeVine oracle could be advantageous in the presence of non-local rewards; the DeVine oracle is insensitive to whether the rewards were applied immediately or at the final time-step of the trajectory, since it only needs the difference of the two trajectory payoffs. On the other hand, all stochastic policy optimization methods (including DDPG and TD3) assume the environment provides an immediate reward at each time-step. The fact that (1) the intermediate time-step rewards are fixed at zero regardless of the actions taken by the agent, and (2) all the actions determine a single reward applied at the final time-step, may impact stochastic policy optimization methods' (i.e., TRPO, PPO, DDPG, and TD3) advantage estimation oracles negatively since their algorithmic design relies on immediate rewards being applied at each time step.
>
> Detailed experiment settings including the specific reward function's definition and the hyper-parameters are described in Section A.9 of the appendix.
>
> **Question 2**: For hyper-parameter optimization, we have performed single seed grid search over a small set of hyper-parameters for DDPG and TD3, which turned out to be futile. To perform this hyper-parameter optimization more efficiently, we manually experimented with many configurations by probing the training curves ourselves and trying to mitigate learning deficiencies by trial and error. We even tried to diagnose and mitigate the divergence behaviors of DDPG and TD3 by making algorithmic amendments such as removing the hyperbolic tan activations at the final layers and using specific final layer initializations and scalings. Our DDPG and TD3 tunning experiments showed a consistent pattern of inability to effectively solve the long-horizon environment. While TD3 and DDPG could perform competitively under short-horizon counter-part environments, they could not effectively solve the long-horizon environment in any of the tested configurations. We have included a more detailed description of the setting and the process in Sections A.9 and A.10 of the appendix.
>
> The poor performance of existing methods under long horizons is well-known in the community and is commonly referred to as the curse of horizon in reinforcement learning ([1]). General worst-case analyses suggests that the sample complexity of reinforcement learning is exponential with respect to the horizon length ([2], [5], and [6]). Deriving polynomial lower-bounds for the sample complexity of reinforcement learning methods is still an open problem ([3]). Lower-bounding the sample complexity of reinforcement learning for long horizons under different settings and simplifying assumptions has been a topic of theoretical research ([4] and [7]). Some recent work has closely examined the importance sampling gradient estimators scaling to horizon in terms of both theoretical and practical estimation variances ([1], [8], and [9]). All in all, long horizons certainly impose scalability challenges for all reinforcement learning methods, especially the ones suffering from excessive estimation variances due to the stochastic policy search procedures employed within them. Our method does not suffer from this curse in the same way because it does not induce exploding variances as the horizon increases.

---

### Decision · Program_Chairs · 2021-01-07
**Final Decision**

**Decision:**

Reject

**Comment:**

This paper proposes a deterministic policy gradient method that does not require to inject noise in the action selection.
Although the reviewers acknowledge that this paper has merits (novel and interesting idea, well written, technically sound), they have some doubts about the motivations for the proposed approach and about its empirical performance: a deeper analysis is requested.
The paper is borderline and needs to be revised before being ready for publication.